The EMBO Journal (2013) 32, 1381–1392
www.embojournal.org

# Transcription-coupled eviction of histones H2A/H2B governs V(D)J recombination

## Sarah Bevington[1] and Joan Boyes*

Institute of Molecular and Cellular Biology, Faculty of Biological Sciences, University of Leeds, Leeds, UK

**Initiation of V(D)J recombination critically relies on the formation of an accessible chromatin structure at recombination signal sequences (RSSs) but how this accessibility is generated is poorly understood. Immunoglobulin light-chain loci normally undergo recombination in pre-B cells. We show here that equipping (earlier) pro-B cells with the increased pre-B-cell levels of just one transcription factor, IRF4, triggers the entire cascade of events leading to premature light-chain recombination. We then used this finding to dissect the critical events that generate RSS accessibility and show that the chromatin modifications previously associated with recombination are insufficient. Instead, we establish that non-coding transcription triggers IgL RSS accessibility and find that the accessibility is transient. Transcription transiently evicts H2A/H2B dimers, releasing 35–40 bp of nucleosomal DNA, and we demonstrate that H2A/H2B loss can explain the RSS accessibility observed _in vivo_. We therefore propose that the transcription-mediated eviction of H2A/H2B dimers is an important mechanism that makes RSSs accessible for the initiation of recombination.**

_The EMBO Journal_ (2013) **32**, 1381–1392. doi:10.1038/emboj.2013.42; Published online 5 March 2013
_Subject Categories:_ chromatin & transcription; immunology
_Keywords:_ chromatin remodelling; gene regulation; non-coding transcription; V(D)J recombination

## Introduction

V(D)J recombination generates a highly diverse set of immunoglobulin and T-cell receptor genes by the regulated joining of individual V, D and J gene segments. The reaction is initiated by two lymphocyte-specific proteins, RAG1 and RAG2, that bind to the conserved recombination signal sequences (RSSs) that flank all recombining gene segments in the immunoglobulin and T-cell receptor loci. Despite using these common elements, recombination is tightly regulated so that the antigen receptor loci rearrange in a strict cell and developmental-stage-specific manner. The key regulator is thought to be the generation of an accessible chromatin

*Corresponding author. Institute of Molecular and Cellular Biology, Faculty of Biological Sciences, University of Leeds, Leeds, West Yorkshire, LS2 9JT, UK. Tel.: +44 113 343 3147; Fax: +44 113 343 3167; E-mail: j.m.boyes@leeds.ac.uk
[1]Present address: School of Immunity and Infection, Institute of Biomedical Research, University of Birmingham, Edgbaston, Birmingham, B15 2TT, UK

structure at the RSSs (Schatz and Ji, 2011). However, the fundamental mechanism by which the RSSs are made accessible for RAG cutting remains poorly understood.

RSS accessibility is generated as part of the highly regulated, multistep process of locus activation for V(D)J recombination. Enhancers play a key role in initiating this process (Cobb _et al_, 2006) and one way in which they do this is by physically contacting promoters of non-coding transcription to activate them (Oestreich _et al_, 2006). The resulting passage of RNA polymerase through the RSSs is essential for recombination (Abarrategui and Krangel, 2009) and has been shown to cause a number of chromatin changes that correlate with recombination, including increased histone acetylation, histone H3 lysine 4 trimethylation (H3K4me3) and histone H3 lysine 36 trimethylation (H3K36me3) (McMurry and Krangel, 2000; Perkins _et al_, 2004; Abarrategui and Krangel, 2006, 2007). Of these, H3K4me3 is essential to recruit RAG2 via its interaction with the RAG2 PHD finger (Liu _et al_, 2007; Matthews _et al_, 2007).

While this modification brings RAG proteins into the vicinity of the RSSs, the packaging of RSSs into nucleosomes (Baumann _et al_, 2003) prevents their subsequent cutting by RAG proteins (Kwon _et al_, 1998; Golding _et al_, 1999; McBlane and Boyes, 2000). Importantly, none of the above transcription-mediated chromatin changes has been shown to directly overcome this nucleosome-mediated repression to make the RSSs accessible (Golding _et al_, 1999; McBlane and Boyes, 2000). Likewise, although the nucleosome remodelling complex, SWI/SNF, is essential for recombination (Osipovich _et al_, 2007), and has been shown to remodel nucleosomes to activate recombination _in vitro_ (Du _et al_, 2008), it remains unresolved whether SWI/SNF performs this function _in vivo_ and/or whether it functions by remodelling promoter nucleosomes to activate non-coding transcription (Osipovich _et al_, 2007; Osipovich _et al_, 2009). Thus, an unanswered question is exactly how the nucleosome-mediated repression is overcome so the RSSs are made available for RAG cutting _in vivo_.

To address this question, and determine the critical events in locus activation that control RSS accessibility, our strategy was to activate V(D)J recombination and then analyse the cascade of events that follow. As a model, we examined the immunoglobulin light-chain (IgL) loci that rearrange during the pre-B-cell stage. Notably, IRF4 binds to enhancers of both light-chain loci, Igκ (κ3'E) and Igλ (Eλ$_{3-1}$/Eλ$_{2-4}$) (Rudin and Storb, 1992; Eisenbeis _et al_, 1993; Schlissel, 2004), and loss of IRF4 blocks B-cell development (Lu _et al_, 2003), correlating with loss of IgL recombination. Since re-introduction of IRF4 into _Irf4−/−_ cells re-activates recombination (Ma _et al_, 2006; Johnson _et al_, 2008), IRF4 is a prime candidate for the activation of IgL recombination.

The levels of IRF4 increase at the pro-B/pre-B transition (Muljo and Schlissel, 2003) and a plausible hypothesis is that this developmentally regulated increase is central to the activation of stage-specific IgL rearrangement. To test this,

we equipped pro-B cells with pre-B-cell levels of IRF4 and probed the impact on recombination, transcription and the chromatin structure of Igκ and Igλ light-chain loci.

We find remarkably that increased levels of IRF4 alone do indeed activate premature light-chain recombination. Importantly, Igλ recombination is activated to pre-B-cell levels, suggesting that increased level of this one factor is enough to trigger the cascade of events, leading to full developmental activation of this locus. In contrast, non-coding transcription and recombination are only partially activated at the Igκ locus. This differential activation then provided us with a unique tool to dissect the critical events that lead to RSS accessibility and recombination. We find strong links between the level of IRF4 and the increase in non-coding transcription as well as between non-coding transcription and RSS accessibility. By investigating mechanistically how non-coding transcription generates RSS accessibility, we show that the RSSs become accessible but only transiently. Transcription transiently evicts H2A/H2B dimers, releasing 35–40 bp of DNA from nucleosomes, and we present mechanistic evidence that this is an important step that can overcome the nucleosome-mediated repression to make RSSs available for recombination. Importantly, since this release is only transient, it allows the initiation of recombination to occur while preventing excessive RAG cutting that could lead to genome instability.

## Results

### Generation of transgenic mice that express increased levels of IRF4 in pro-B cells

To specifically increase the level of IRF4 (also known as PIP) in pro-B cells, its cDNA was cloned under the control of the pro-B-cell-specific promoter and locus control region from the λ5 locus (Figure 1A) (Sabbattini et al, 1999). Three transgenic lines were generated (PIP2, PIP3 and PIP4) with transgene copy numbers of 16, 16 and 5, respectively (Supplementary Figure S1A) and the levels of IRF4 were examined in pro-B cells (CD43 + /CD19 + ) purified from primary bone marrow cultures. Notably, we find that in pro-B cells from the PIP3 transgenic line, IRF4 is expressed at the same level as in pre-B cells at both the protein (Figure 1B, left) and mRNA levels (Supplementary Figure S1B). IRF4 expression is also increased in the two other transgenic lines, PIP4 and PIP2, at 2.1- and 1.4-fold higher, respectively, than that found in pro-B cells of non-transgenic mice (NTG) (Figure 1B, right).

### Increased IRF4 levels trigger premature light-chain recombination in pro-B cells

Having equipped pro-B cells in the PIP3 transgenic line with pre-B-cell levels of IRF4, we next asked if this is sufficient to reprogramme stage-specific IgL rearrangement. In NTG, light-chain recombination primarily occurs in pre-B cells with only 15% of cells rearranging the Igκ locus at the earlier pro-B (CD43 + /CD19 + )-cell stage (Novobrantseva et al, 1999). To determine if increased levels of IRF4 in pro-B cells trigger premature rearrangement, we purified primary pro-B cells from PIP transgenic as well as from NTG by flow cytometry and analysed the levels of recombination.

We find remarkably that increased IRF4 levels alone indeed cause increased Igκ recombination by about two-fold (Figure 2A). Consistent with this, the recombined kappa

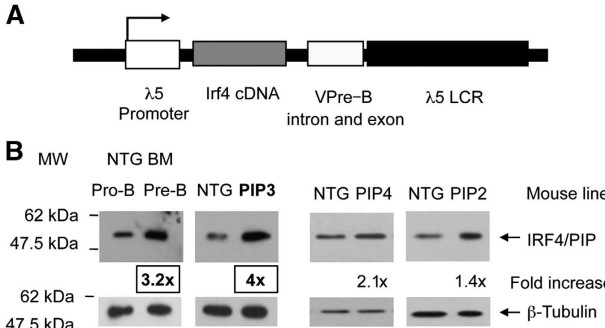

**Figure 1** Increased IRF4 levels in pro-B cells from three transgenic lines. (**A**) Diagram of the λ5/V_{pre-B} construct. (**B**) Left: IRF4 protein levels in purified primary pro-B and pre-B cells (NTG BM). Right: IRF4 levels in cultured pro-B cells from the PIP3, PIP4 and PIP2 lines compared to cultured pro-B cells from NTG. Protein loading was normalised to β-tubulin.

light-chain cell surface marker, Igκ, is expressed by approximately twice as many pro-B cells from the PIP3 mice compared with those from NTG (Supplementary Figure S2A, left). Moreover, control flow cytometry analyses showed that other pro-B-cell markers are unchanged (Supplementary Figure S3), suggesting that the increase in recombination is not due to differentiation of the cells. Despite the marked increase in Igκ recombination, increased IRF4 levels do not completely activate Igκ recombination, which normally increases seven-fold in pre-B cells.

One possible reason for this incomplete activation might be because the increased levels of IRF4 alone are insufficient to trigger Igκ locus contraction that brings the V and J gene segments into close proximity for recombination (Roldan et al, 2005). However, Vκ21G and Jκ1 lie only 18.4 kb apart, and recombination between these gene segments could be achieved by chromatin looping that is frequently observed at non-antigen receptor loci, rather than by large-scale locus contraction. Nevertheless, Vκ21/Jκ1 recombination is also increased by only about two-fold in the presence of increased IRF4 (Supplementary Figure S2A right). These data therefore imply that IRF4 positively activates recombination but, as described in other studies (Johnson et al, 2008; Malin et al, 2010) and below, additional factors, other than locus contraction, prevent full upregulation of Igκ rearrangement in pro-B cells.

We next examined Igλ recombination and find an exceptionally large increase in pro-B cells from the PIP3 mice compared to those from NTG (Figure 2B). Indeed, the level of recombination is the same as that normally observed in pre-B cells. Since IRF4 levels are at physiological, pre-B-cell levels in this transgenic line, this suggests that an increase in this single transcription factor is enough to completely reprogramme developmental activation of Igλ recombination. Consistent with this, the level of mature Igλ transcripts in pro-B cells from the PIP3 transgenic mice is also increased to the same level as in pre-B cells (Supplementary Figure S2B). Furthermore, smaller increases in the level of Igλ recombination are observed in the PIP4 and PIP2 transgenic lines, correlating with the lower levels of IRF4 (Figure 2B and C).

### Increased IRF4 levels induce non-coding transcription at the IgL loci

Non-coding transcription increases at the same developmental stage as the initiation of light-chain recombination

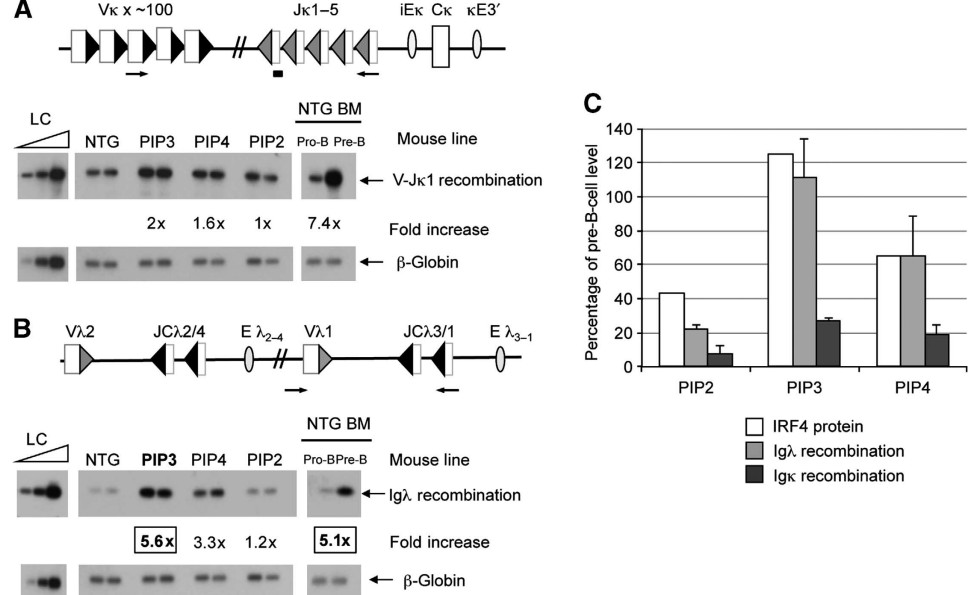

**Figure 2** Increased IRF4 levels in pro-B cells cause premature light-chain recombination. (**A, B**) Upper: Schematics of the *Ig*κ and *Ig*λ loci. Arrows indicate the locations of primers used in the PCR assay, filled triangles represent RSSs; Eλ$_{2-4}$/Eλ$_{3-1}$ indicate the *Ig*λ recombination enhancers; κ3′E, the *Ig*κ 3′ enhancer; κEi, the *Ig*κ intronic enhancer and V, J and C indicate variable, joining and constant regions, respectively. The probe to detect *Ig*κ recombination is shown as a black bar. (**A, B**) Lower: Southern blots showing the level of (**A**) *Ig*κ and (**B**) *Ig*λ recombination products in cultured pro-B cells from the PIP3, PIP4 and PIP2 transgenic lines and NTG and pro-B cells and pre-B cells that had been purified directly from the bone marrow of NTG (NTG BM). Duplicate samples were analysed for each transgenic line. The fold increase is compared to cultured pro-B cells from NTG. The amounts of DNA were normalised to a region of the β-*globin* gene. LC is the loading control, DNA was added in three-fold increments. (**C**) Graph showing the relative IRF4, *Ig*λ and *Ig*κ recombination levels in the transgenic lines. The recombination data are an average of three experiments; s.d. is shown. Source data for this figure is available on the online supplementary information page.

(Engel *et al*, 1999) and, having established that IRF4 can trigger IgL recombination, we next asked if it does this by activating non-coding transcription.

We find that non-coding transcription of the *Ig*κ JC and Vκ02 regions is noticeably increased in pro-B cells from the PIP3 transgenic line (Figure 3A and Supplementary Figure S4). Importantly, the increase in transcription correlates extremely well with the increase in *Ig*κ recombination: recombination is increased to 30% of the pre-B-cell level (Figure 2A) whereas *Ig*κ JC and Vκ02 transcription are stimulated to 26 and 39% of pre-B-cell levels, respectively.

Next, we examined non-coding transcription through the *Ig*λ gene segments and find it is substantially increased in response to increased IRF4 (Figure 3B and Supplementary Figure S4). Indeed, non-coding transcription through both the Jλ$_1$ and Vλ$_{1/2}$ genes in pro-B cells from the PIP3 line is close to the level in pre-B cells. Moreover, consistent with the idea that IRF4 directly regulates *Ig*λ activation, we see smaller increases in non-coding transcription in the PIP4 and PIP2 transgenic lines. Together, these data suggest that increased IRF4 levels directly reprogramme IgL non-coding transcription and recombination, and that IgL transcription and recombination are extremely tightly linked.

### Increased IRF4 triggers H3K4me3 at the J gene segment RSSs

Transcription is known to play a key role in regulating V(D)J recombination (Abarrategui and Krangel, 2007; Xu and Feeney, 2009) but exactly how it does this is unknown. Transcription increases various chromatin modifications that correlate with recombination, and of these, H3K4me3 is required for the recruitment of RAG2 (Liu *et al*, 2007;

Matthews *et al*, 2007). However, which, if any, of the other chromatin changes plays a role in the activation of recombination and/or the generation of RSS accessibility is poorly understood. Importantly, since IRF4 activates the *Ig*κ and *Ig*λ loci to different extents, this provided us with a powerful tool to investigate which transcription-mediated chromatin changes are tightly linked to recombination to thus gain an insight into which modification(s) are (a) important and (b) potentially prime triggers of the reaction.

First, we examined H3K4me3 and find that this modification is substantially increased at both IgL loci between the pro-B- and pre-B-cell stages (Figure 4). Moreover, this change is fully reproduced by the increased IRF4 levels in PIP3 mice.

Notably, however, although H3K4me3 is increased in response to IRF4, we find that in pro-B cells from the PIP3 line, the level of H3K4me3 at the *J*κ RSSs is even higher than that in pre-B cells (Figure 4, right). Despite this, *Ig*κ recombination is only partially activated. This indicates that H3K4me3 and recombination are not tightly linked and, since H3K4me3 causes the recruitment of RAG2 (Ji *et al*, 2010b), these data imply that recruitment of RAG2 is not sufficient for full activation of *Ig*κ recombination.

### The RSSs at the Igκ and Igλ loci show different levels of accessibility

The generation of an accessible chromatin structure at RSSs is essential for recombination and we next asked if this is more tightly linked with recombination and thus if it is potentially the prime trigger. To this end, restriction enzyme sites close to the RSSs were identified and used to directly measure accessibility in nuclei from pro- and pre-B cells from NTG as well as pro-B cells from the PIP3 transgenic mice (Figure 5A).

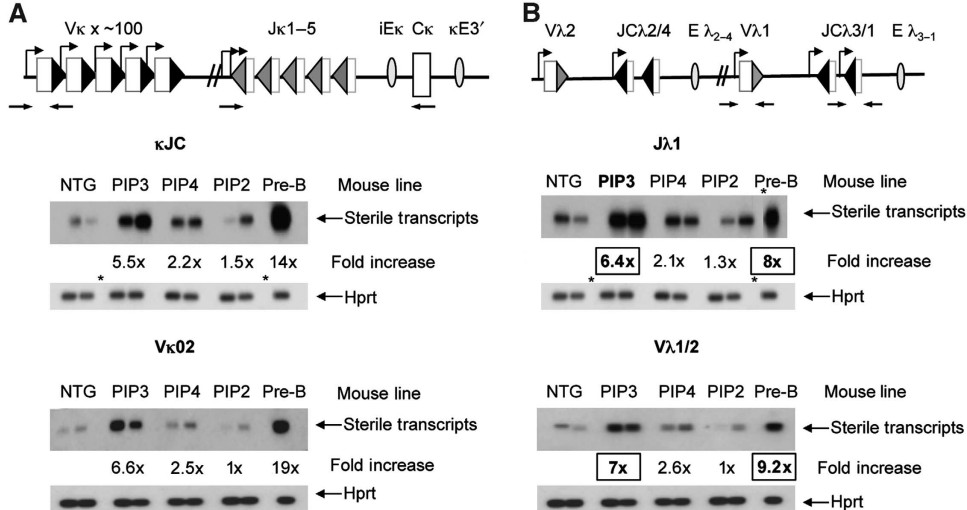

**Figure 3** Increased IRF4 levels in pro-B cells cause premature non-coding transcription of the *Igκ* and *Igλ* loci. (**A, B**) Upper: schematic of the *Igκ* and *Igλ* loci. Labels are as for Figure 2. Lower: Southern blots showing the level of non-coding transcripts at (**A**) the *Igκ* locus through the *κJC* and *Vκ02* regions, and (**B**) at the *Igλ* locus through the *Vλ1/2* and *Jλ1* gene segments. For the *V* genes, a probe was used that detects transcripts from both *Vλ1* and *Vλ2*. Transcripts were measured in pro-B cells from transgenic mice, NTG and from pre-B cells from NTG. The fold increase shown is compared to pro-B cells from NTG. The amounts of cDNA were normalised to *Hprt* transcripts; duplicate samples are shown from each transgenic line. The asterisks indicate where lanes have been removed from a continuous gel. Source data for this figure is available on the online supplementary information page.

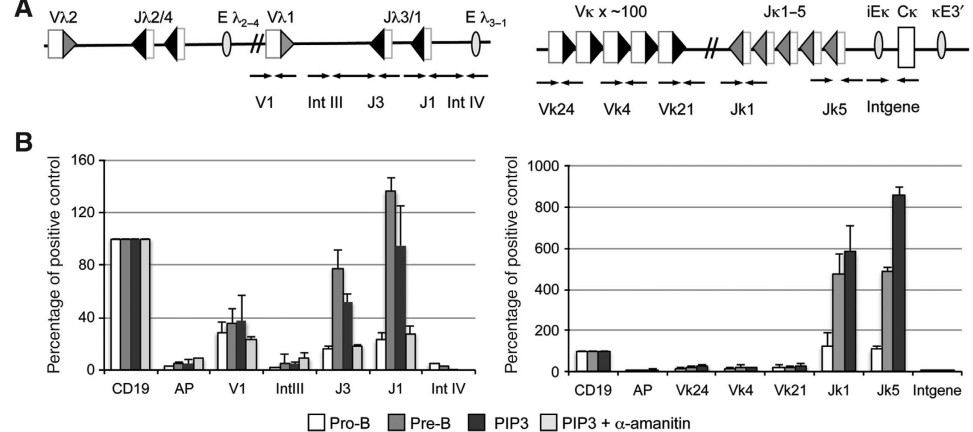

**Figure 4** Increased IRF4 levels in pro-B cells stimulate H3K4me3 at the *Igκ* and *Igλ* loci. (**A**) Schematic of the *Igλ* and *Igκ* loci. Labels are as for Figure 2. (**B**) H3K4me3 levels at the *Igλ* (left) and the *Igκ* (right) loci in primary pro-B cells and pre-B cells from NTG and pro-B cells from PIP3 transgenic mice. The levels of H3K4me3 are shown as a percentage of the positive control, the CD19 promoter; the negative control (AP) is the α-amylase promoter. Int III, Int IV and Intgene are intergenic regions. The s.d. from three independent experiments is shown.

We find that accessibility is markedly increased in pre-B cells compared to pro-B cells at the $Jλ_1$ and $Jκ_3$ RSSs in NTG: accessibility is less than 4% in pro-B cells, compared to 16–18% in pre-B cells (Figure 5A, left). Importantly, increased IRF4 levels in pro-B cells from the PIP3 mice result in a dramatic increase in accessibility at the $Jλ_1$ gene segment that is very close to the accessibility in pre-B cells (Figure 5A, left); this correlates extremely well with increased *Igλ* recombination. Likewise, at the *Igκ* locus, where recombination is increased by only two-fold, RSS accessibility is only marginally increased (Figure 5A, right). Thus, IRF4 causes increased RSS accessibility and there is an extremely tight correlation between RSS accessibility and the level of *Igκ* and *Igλ* recombination.

Next, to investigate if transcription regulates this change in RSS accessibility, we measured accessibility when transcription is ongoing and following its inhibition with α-amanitin. At both $Jκ_3$ and $Jλ_1$, accessibility is high in pre-B cells

(16–18%). Upon inhibition of transcription, cutting is substantially decreased to close to pro-B-cell levels (~5%; Figure 5A). Likewise, at the $Jλ_3$ and $Vλ_1$ gene segments, the high level of accessibility in pre-B cells is lost upon inhibition of transcription (Supplementary Figure S5). These data therefore strongly imply that non-coding transcription plays a critical role in generating RSS accessibility.

### A new chromatin change generates RSS accessibility

The extremely tight link between RSS accessibility and recombination (Figures 2 and 5) suggests that the generation of RSS accessibility could be the prime trigger of recombination. Therefore, to determine how transcription generates RSS accessibility, we used the differential activation of the *Igκ* and *Igλ* loci to investigate which of the transcription-mediated chromatin changes correlate strongly with recombination and thus which might ultimately trigger initiation of the reaction.

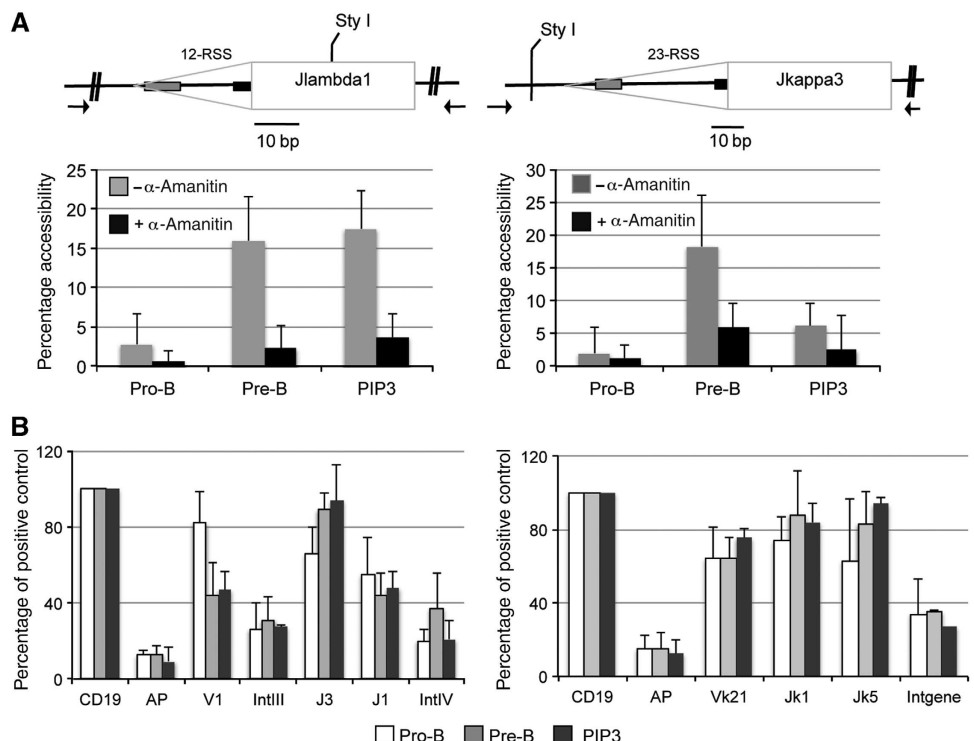

**Figure 5** Non-coding transcription increases accessibility at RSSs in pre-B cells. (**A**) Upper: schematics of the $J\lambda_1$ (left) and the $J\kappa_3$ (right) gene segments and RSSs. The grey triangle represents the RSS and black and grey boxes represent the heptamer and nonamer, respectively. The restriction sites used to probe accessibility are shown. Arrows show the positions of the PCR primers. Lower: accessibility of the $J\lambda_1$ and $J\kappa_3$ gene segments in primary pro-B and pre-B cells from NTG and pro-B cells from the PIP3 transgenic line. Error bars show the s.d. from at least three experiments. (**B**) Histone H4 acetylation levels at the $Ig\lambda$ locus (left) and the $Ig\kappa$ locus (right) in primary pro-B and pre-B cells from NTG and pro-B cells from PIP3 transgenic mice. The levels of acetylation are shown as a percentage of the positive control, the CD19 promoter. The s.d. from three independent experiments is shown.

Histone acetylation has been linked with increased locus-wide accessibility (Hebbes et al, 1994) and, since acetyltransferases associate with elongating RNA polymerase II (Wittschieben et al, 2000), this seemed a potential candidate for increasing RSS accessibility. Therefore, we examined the levels of histone H4 acetylation using an antibody that recognises all five acetylated N-terminal lysine residues. Surprisingly, at both $Ig\lambda$ and $Ig\kappa$ loci, histone H4 acetylation at the RSSs is high in pro-B and does not increase following the transition to pre-B cells (Figure 5B). Consistent with this, we find high levels of histone H4 acetylation in pro-B cells from the PIP3 transgenic line at the RSSs of both light-chain loci. Thus, histone H4 hyper-acetylation appears to be an early step in the activation of the light-chain loci and transcription does not appear to generate RSS accessibility via this modification.

Likewise, similar to previous studies (Goldmit et al, 2005), the level of H3 acetylation is increased by only a small amount (maximum two-fold) at both light-chain loci between pro-B and pre-B cells (Supplementary Figure S6A). Therefore, neither H4 nor H3 acetylation appears to be sufficient to increase RSS accessibility, but instead, acetylation at RSSs might be a pre-requisite for other chromatin changes.

Transcription is also known to increase H3K36me3 and this correlates well with recombination at the TCRα locus (Abarrategui and Krangel, 2007). However, we find no such correlation at the $Ig\lambda$ locus (Supplementary Figure S6B), suggesting that this modification too is unlikely to be the critical change that generates RSS accessibility.

Together, these data imply that none of the chromatin changes previously linked with recombination (H3K4me3, H3K36me3 and histone acetylation) correlate with increased RSS accessibility at the IgL loci. Moreover, none of the other transcription-mediated histone modifications has been shown to directly increase the accessibility of DNA in nucleosomes. Instead, these data raise the possibility that something intrinsic to the process of transcription itself remodels nucleosomes to generate RSS accessibility.

### Eviction of an H2A/H2B dimer increases RSS accessibility and RAG cutting

Transcription is known to evict an H2A/H2B dimer from a nucleosome onto the chaperone FACT, temporarily converting the nucleosome into a hexasome (a nucleosome lacking an H2A/H2B dimer) (Kimura and Cook, 2001; Kireeva et al, 2002; Thiriet and Hayes, 2005). This reduces histone/DNA contacts and could potentially increase RSS accessibility in two ways: first, for the region of DNA that remains associated with histones, there are fewer histone/DNA contacts in the hexasome compared to a nucleosome and this could potentially permit RAG binding to the RSS. Second, transcription-mediated eviction of the distal H2A/H2B dimer (Kulaeva et al, 2009) transiently releases about 35–40 bp of nucleosomal DNA. Thus, if an RSS lies within this 35 bp of DNA, it would become totally accessible for RAG cutting.

To test if the formation of a hexasome indeed facilitates RAG cutting, we used three DNA templates where the RSS is

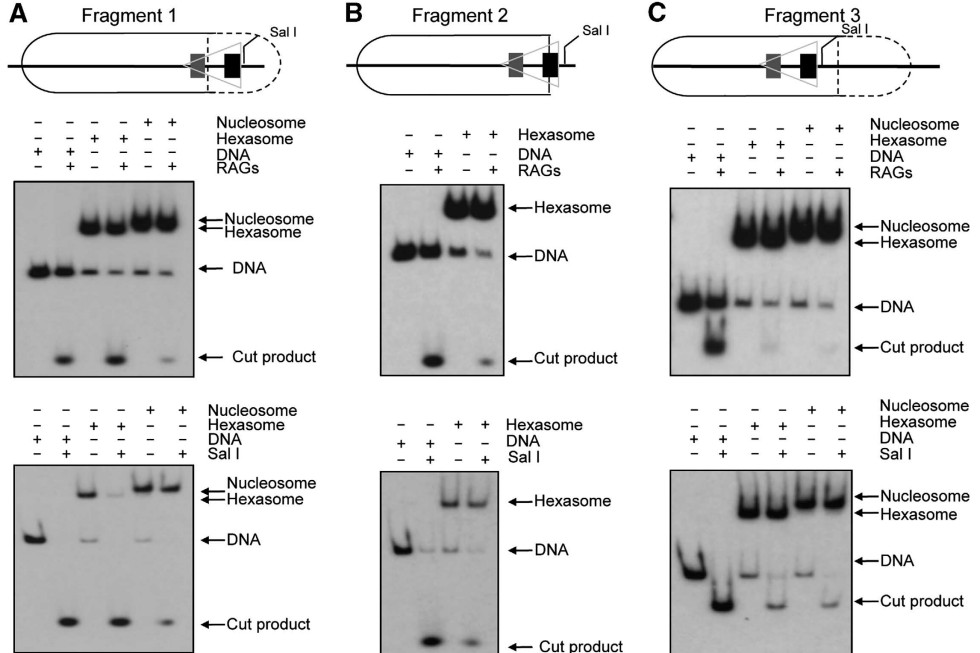

**Figure 6** Eviction of an H2A/H2B dimer makes RSSs accessible for RAG cutting. Schematic (upper panel), RAG cutting (middle panel) and Sal I cutting (lower panel) of (**A**) free DNA, hexasome and nucleosome where the heptamer and the RSS spacer are not protected by histones in the hexasome, (**B**) free DNA and hexasome where the heptamer of the RSS is not protected by histones and (**C**) reconstitutes where the nucleosome and hexasome protect the RSS. The solid lines in the schematic show the region protected by the hexasome; the dotted lines show the additional region protected by the nucleosome (Supplementary Figure S7). In (**C**), the amount of DNA in the nucleosome and hexasome preparations is identical. The grey triangle represents the RSS and the heptamer and nonamer of the RSS are represented by black and grey boxes, respectively.

located at different positions along the DNA sequence. These were reconstituted into either a nucleosome or a hexasome *in vitro* with recombinant *Xenopus* histones (Figure 6), and following gel isolation, micrococcal nuclease digestion confirmed that 147 bp of DNA was protected by the nucleosome core particles whereas only 105–110 bp were protected by the hexasome (Supplementary Figure S7B). Mapping experiments then confirmed that for Fragment 1 both the heptamer and 12-spacer are exposed; for Fragment 2, only the heptamer of the RSS is exposed whereas for Fragment 3 the entire RSS was protected by histones (Supplementary Figure S7B).

RAG cutting was then analysed using manganese as the divalent cation. Although under these conditions RAGs perform uncoupled cleavage, which differs from the coupled cleavage that occurs *in vivo* (van Gent *et al*, 1996); this did enable us to analyse the ability of RAG proteins to cleave the individual reconstitutes. We find strikingly that on the hexasome where the RSS-heptamer plus 12-spacer are exposed, there is a very high level of RAG cutting that is equivalent to that on free DNA (Fragment 1; Figure 6A). Likewise, using a restriction enzyme as a probe, accessibility of a Sal I site, that lies within 2 bp of the heptamer, is also comparable to that on free DNA (Figure 6, lower).

Next, we examined RAG cutting on hexasomes reconstituted onto Fragment 2. Here, where only the heptamer of the RSS is exposed, we find a more modest level of cutting that is reflected by a similar, moderate amount of cutting by Sal I (Figure 6B). Finally, for reconstitutes made with Fragment 3, there is negligible RAG cutting on both the hexasome and the nucleosome (Figure 6C). These data therefore suggest that when the RSS remains protected by histones (Fragment 3), RAG cutting is inhibited. However, if the RSS lies within the

35–40 bp of DNA that is released upon formation of the hexasome, high-level RAG cutting can occur to thus enable the initiation of V(D)J recombination. Our data also confirm that restriction enzyme cutting correlates very well with the level of RAG cutting and RSS accessibility, implying that these enzymes are valid probes for RSS accessibility.

### RSS accessibility is transient and stochastic

The above experiments imply that eviction of H2A/H2B dimers by non-coding transcription can expose the RSS to allow RAG cutting. *In vivo*, H2A/H2B eviction is known to be transient with a half-life of about 6 min (Kimura and Cook, 2001). Thus, if our hypothesis is correct, the accessibility of RSSs *in vivo* is expected to be transient. Therefore, we inhibited transcription with α-amanitin and measured RSS accessibility with time: RSS accessibility is completely lost within 15 min of transcription inhibition (Figure 7A). These experiments thus imply that RSS accessibility is indeed transient and is on a time scale entirely consistent with that expected from the eviction of H2A/H2B dimers. Moreover, control experiments showed that addition of α-amanitin does not result in loss of histone acetylation, which is another potentially short-lived chromatin change. Notably, since RSS accessibility correlates tightly with the level of recombination, these data imply that H2A/H2B eviction could be an important trigger that enables the initiation of recombination.

To further test this hypothesis, we compared the levels of H2B and H3 at the Jκ RSSs *in vivo*. Consistent with the idea that the H2A/H2B dimer is lost in a transcription-dependent manner, we see an average 25% decrease in H2B compared to H3 at the transcribed Jκ RSSs (Figure 7B). Moreover, addition of α-amanitin eliminated this reduction of H2B at the Jκ RSSs

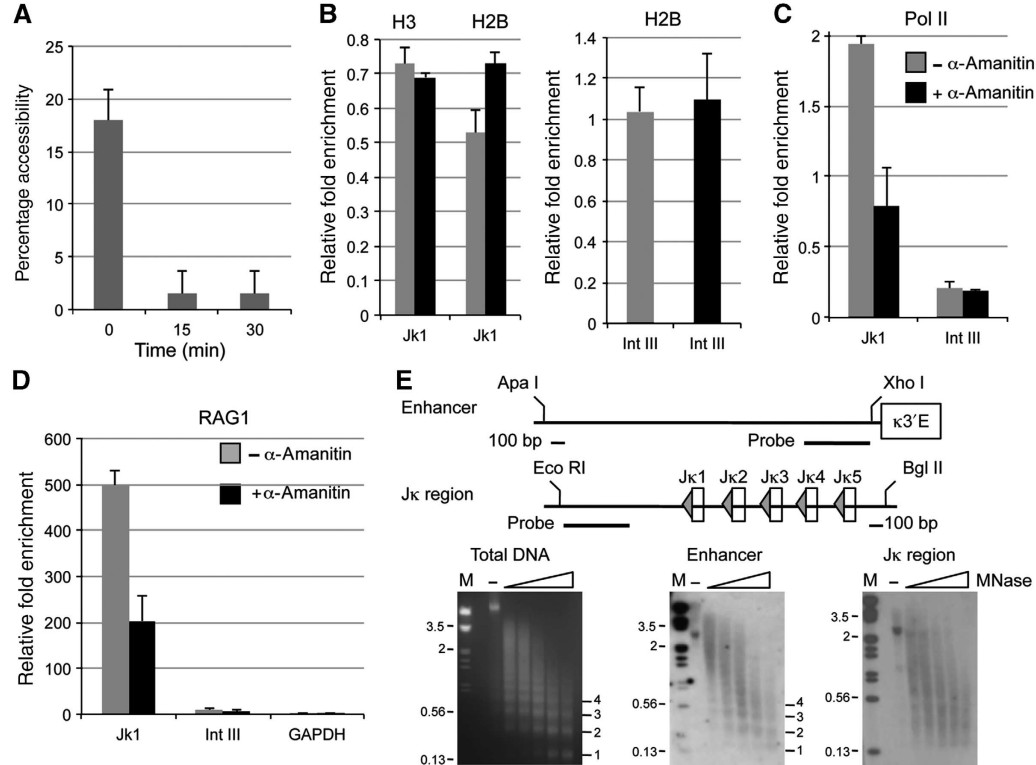

**Figure 7** RSSs are transiently and stochastically made accessible via eviction of H2A/H2B dimers. (**A**) Accessibility at the RSSs is transient. Nuclei were isolated from pre-B cells and accessibility at the $J\lambda_1$ Sty I site measured at the times shown following addition of α-amanitin. The data are the average of two experiments. (**B**) Transcription-dependent eviction of H2B at the $J\kappa1$ RSS. The levels of H2B and H3 were compared in primary pre-B cells at the $J\kappa1$ RSS in the presence (black bars) or absence (grey bars) of α-amanitin (left). Changes in H2B in the presence and absence of α-amanitin at a region not known to be transcribed (Intgene III) are shown on the right. Values at $J\kappa1$ were normalised to those at a non-transcribed region; the values at the Int III region are averaged values using normalised amounts of DNA. The upstream primer is in a region that is deleted upon recombination; thus the changes are measured in pre-B cells that have not yet undergone recombination. The data are an average of five experiments; error bars show s.d. (**C**) Eviction of H2B correlates with the presence of RNA polymerase II. The presence of RNA polymerase II as determined by chromatin immunoprecipitation at the $J\kappa1$ RSS and at the non-transcribed Intgene III region in the presence and absence of α-amanitin. Values were normalised to a non-transcribed region (Int V); the average of three experiments with s.d. is shown. (**D**) Eviction of H2B correlates with increased RAG1 binding. Binding was examined at the $J\kappa1$ RSS, the Intgene III region and the *Gapdh* gene as an example of a transcribed region. Numbers were normalised to *Gapdh* or, in the case of *Gapdh*, by using equivalent amounts of DNA. The data are the average of three experiments; s.e. is shown. (**E**) Nucleosome mapping at the $J\kappa$ region and adjacent to κ3′E. Nuclei were isolated from 103/BCL-2 cells that had been grown at 39.7°C to induce IgL recombination and digested with increasing amounts of micrococcal nuclease. The isolated DNA was digested with the restriction enzymes shown and the probes used in indirect end-labelling are indicated (black bars). The region adjacent to κ3′E (middle panel) is shown as a control since promoters and enhancers are thought to promote positioning of adjacent nucleosomes, generating a characteristic nucleosome ladder. A more smeared pattern is apparent at the $J\kappa$ RSSs (right).

but not at an intergenic region that is not known to be transcribed (Figure 7B). A similar transcription-dependent loss of H2A was also observed at the $J\kappa$ RSSs but not at a non-transcribed region (Supplementary Figure S8A).

To further determine if H2A/H2B eviction is indeed dependent on ongoing transcription, we examined the association of the serine 2 phosphorylated form of RNA polymerase II by chromatin immunoprecipitation. We find that the elongating form of RNA polymerase II is present at the $J\kappa$ region and that this association is reduced upon addition of α-amanitin. Notably, the elevated levels of RNA polymerase II inversely correlate with the presence of H2A/H2B (Figures 7B and C and Supplementary Figure S8A), thus reinforcing the idea that H2A/H2B eviction is indeed mediated by transcription.

Next, to determine if loss of H2A/H2B facilitates RAG binding, we performed chromatin immunoprecipitation using an anti-RAG1 antibody. RAG2 has been shown to bind at numerous sites in the genome that are enriched in H3K4me3 and thus is unlikely to be tightly linked to ongoing transcription (Ji *et al*, 2010b); in contrast, the binding of

RAG1 has been proposed to be transcription-dependent (Ji *et al*, 2010a). Consistent with the idea that transcription-mediated eviction of H2A/H2B dimers facilitates RAG binding to the RSSs, we observe an increase in RAG1 binding at $J\kappa1$ when transcription is ongoing, that is significantly reduced by addition of α-amanitin (Figure 7D). However, RAG1 association appears more complex than just binding to RSSs: some RAG1 persists in the absence of transcription. This could occur in various ways, including via interaction with RAG2, and we propose such local retention of RAG1 could be important to facilitate RAG binding when the RSSs do become transiently accessible. Notably, RAG1 binding is not observed at an intergenic region nor at a transcribed gene that lacks an RSS (Figure 7D). Together, these data imply that transcription-mediated eviction of H2A/H2B dimers facilitates RAG binding to RSSs to initiate V(D)J recombination.

A further prediction of our model is that where the RSS lies within the nucleosome will critically determine whether or not the RSS becomes accessible upon H2A/H2B eviction. Previously, we found that RSSs are preferentially protected

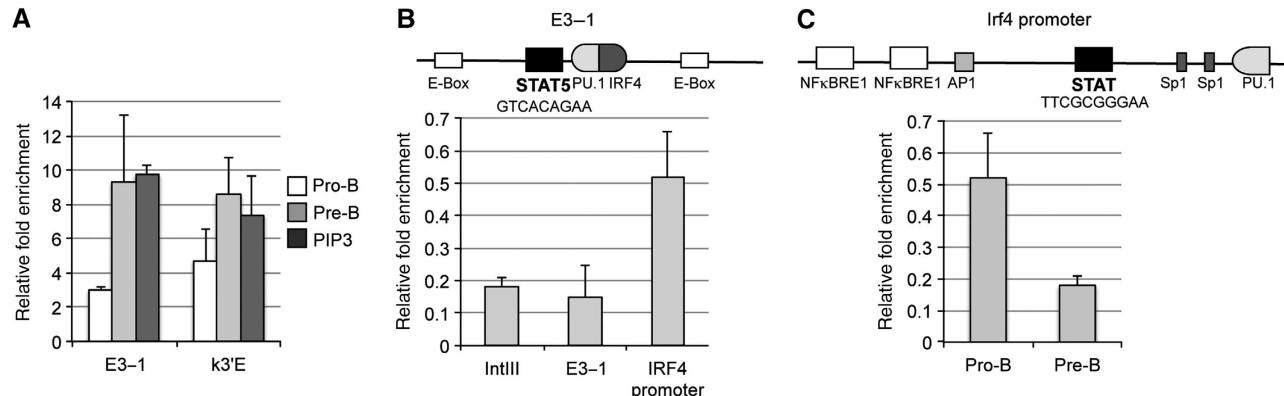

**Figure 8** Differences in STAT5-mediated repression of $Ig\lambda$ and $Ig\kappa$ recombination. (**A**) Chromatin immunoprecipitation of IRF4 at the $Ig\lambda$ (E3-1) and $Ig\kappa$ (k3'E) recombination enhancers in pro-B cells and pre-B cells from NTG and pro-B cells from the PIP3 transgenic mice. The data are an average of at least three experiments; s.d. is shown. (**B**) STAT5 does not bind to the $E\lambda_{3-1}$ enhancer (E3–1) but is enriched at the $Irf4$ promoter in pro-B cells. (**C**) Binding of STAT5 to the $Irf4$ promoter decreases at the pro-B/pre-B transition.

by a nucleosome (Baumann *et al*, 2003) but these experiments did not distinguish where the RSS lies with respect to the histone octamer. We therefore performed micrococcal nuclease digestion and indirect end labelling, and found that nucleosomes occupy different translational positions at both the $Ig\kappa$ and $Ig\lambda$ J regions (Figure 7E and data not shown). This suggests that upon eviction of H2A/H2B and exposure of 35–40 bp of DNA, the RSS will become accessible in some cells while in others it will remain protected. In turn, this implies that H2A/H2B eviction will promote the stochastic use of different RSSs and thus the generation of a diverse antigen receptor repertoire.

### Differences in STAT5 repression explain differences in *Igκ* and *Igλ* recombination

The preceding data explain mechanistically how non-coding transcription leads to the differential RSS accessibility at the $Ig\kappa$ and $Ig\lambda$ loci. However, these data do not explain why IRF4 activates $Ig\kappa$ and $Ig\lambda$ non-coding transcription to different levels. Since binding sites for IRF4 are present at $\kappa3'E$ (Schlissel, 2004) and the lambda recombination enhancers, $E\lambda_{3-1}$ and $E\lambda_{2-4}$ (Eisenbeis *et al*, 1995), one possibility is that differences in IRF4 binding to the recombination enhancers result in differential $Ig\kappa$ and $Ig\lambda$ activation. However, this does not seem to be the case since IRF4 binding to both enhancers in pro-B cells from the PIP3 transgenic mice is equivalent to that in pre-B cells, and higher than that in pro-B cells from NTG (Figure 8A).

IL-7 represses recombination of both IgL chain loci in pro-B cells via STAT5 (Johnson *et al*, 2008; Malin *et al*, 2010), and we next investigated if this repression modulates the differential IgL recombination. STAT5 represses $Ig\kappa$ recombination by binding to the second recombination enhancer, iE$\kappa$ (Malin *et al*, 2010). Although we find no detectable binding of STAT5 to $E\lambda_{3-1}$ in pro-B cells (Figure 8B), it is bound at the $Irf4$ promoter in pro-B cells but not in pre-B cells (Figure 8C). From this, we suggest that the differential activation of the IgL loci by IRF4 can be explained by the different ways STAT5 represses these loci: STAT5 directly represses $Ig\kappa$ via iE$\kappa$; since IRF4 does not bind to this enhancer, this repression is not overcome by increased IRF4 levels in the PIP3 mice. Binding of IRF4 to $\kappa3'E$ causes the observed small increase in $Ig\kappa$ recombination. In contrast, STAT5 indirectly represses $Ig\lambda$

recombination by binding to the $Irf4$ promoter in pro-B cells. In the PIP3 mice, where $Irf4$ is expressed from the $\lambda5$ promoter, this repression is by-passed and IRF4 binding to $E\lambda_{3-1}$ leads to full $Ig\lambda$ activation.

## Discussion

The generation of an accessible chromatin structure at RSSs is central to the regulation of V(D)J recombination. We report here that remarkably an increase in the level of a single transcription factor is sufficient to trigger the cascade of events leading to RSS accessibility and IgL locus recombination. The fact that activation of the $Ig\kappa$ and $Ig\lambda$ loci was not equivalent provided us with a powerful system to dissect the critical events that regulate RSS accessibility. From this, we propose that the non-coding transcription triggers the transient eviction of H2A/H2B dimers and that this is an important step that enables the initiation of recombination.

The finding that increased levels of IRF4 alone completely activate $Ig\lambda$ recombination in pro-B cells is striking and implies that IRF4 acts very early in the multistep process of locus activation. Indeed, ChIP analyses showed that IRF4 is bound at $\kappa3'E$ and $E\lambda_{3-1}$ enhancers in pre-B cells (Figure 8A) but not elsewhere in the $Ig\lambda$ locus. Since there is also a good correlation between the levels of IRF4 and the increases in $V\lambda1/2$ and $J\lambda_1$ non-coding transcription in the PIP2, PIP3 and PIP4 transgenic mice, this suggests that IRF4 plays a direct role in activating non-coding transcription. A likely mechanism is via its binding to the recombination enhancers that then activate the promoters of non-coding transcription.

Non-coding transcription through the RSSs is essential for recombination (Abarrategui and Krangel, 2009), and our studies demonstrate an extremely tight link between the upregulation of non-coding transcription and recombination at both the $Ig\kappa$ and $Ig\lambda$ loci. However, exactly how non-coding transcription activates recombination was unknown. Previous studies showed that inhibition of non-coding transcription caused loss of various chromatin modifications, including histone acetylation, H3K4me3 and H3K36me3 (Abarrategui and Krangel, 2007). But, these studies did not show which, if any, of these modifications are required for recombination, nor did they show what is the key event that triggers recombination. By performing a comparative

analysis of the $Ig\kappa$ and $Ig\lambda$ loci in pro-B cells from the PIP3 mice, we addressed these fundamental questions and now provide a new mechanistic explanation of how non-coding transcription can activate recombination.

First, by analysing histone acetylation in primary pro- and pre-B cells, we find, as noted previously (Goldmit *et al*, 2005), that acetylation is already elevated at both the $Ig\kappa$ and $Ig\lambda$ loci in pro-B cells. This implies that increased non-coding transcription is not essential for histone hyper-acetylation at these loci; it also indicates that acetylation is an early event in locus activation. Moreover, consistent with previous observations (Johnson *et al*, 2003), acetylation is found preferentially at the RSSs. How the acetylation mark is targeted to the RSS is unknown but since histone H3 acetylation can recruit at least one H3K4 methyltransferase (Nightingale *et al*, 2007b) it is possible that acetylation primes the RSSs for the addition of subsequent activating marks. Consistent with this, we find that H3K4me3 increases at the $J\lambda$ and $J\kappa$ RSSs in pre-B cells but is low in the intergenic regions, similar to the pattern of histone H3 hyperacetylation.

Non-coding transcription appears to directly increase H3K4me3 since inhibition of transcription with α-amanitin results in loss of H3K4me3 at RSSs (Figure 4B, left). Consistent with this, SET1 methyltransferase, which catalyses methylation of H3K4, is associated with the serine 5 phosphorylated form of RNA polymerase II (Li *et al*, 2007). However, at both light-chain loci, H3K4me3 is increased only at the $J$ gene segments and not the $V$ gene RSSs even though transcription is enhanced at both these regions in pre-B cells (Figure 4). One possible explanation for this is the distance of the RSSs from the promoters of non-coding transcription: the $J$ gene RSSs are immediately adjacent to the promoter whereas the $V$ gene RSSs are at least 700 bp away and thus more distant from the highest density of SET1. Although the exact mechanism by which H3K4me3 is preferentially deposited at $J$ gene segments is unknown, our data agree well with recent studies that showed increased H3K4me3 and RAG binding occur predominantly at $J$ gene segments (Ji *et al*, 2010b). It is also notable that the level of H3K4me3 at the $J\kappa$ RSSs is significantly higher (by ~4.5-fold) than at the lambda locus (Figure 4B). Since the murine $Ig\kappa$ locus rearranges 20-fold more frequently than the $Ig\lambda$ locus, the higher level of H3K4me3 could increase RAG recruitment, thereby contributing to the difference in recombination of the two loci.

Importantly, even when high levels of histone acetylation and H3K4me3 are present at the $Ig\kappa$ locus, recombination is not fully activated. Moreover, we find that H3K36me3 remains at the same low level between pro- and pre-B cells at the $Ig\lambda$ locus (Supplementary Figure S6B), suggesting that this modification also is not a prime trigger for recombination. These are the only chromatin modifications that have been correlated with V(D)J recombination (Abarrategui and Krangel, 2007; Liu *et al*, 2007; Matthews *et al*, 2007; Nightingale *et al*, 2007a) and none directly increase RAG access to RSSs although they can recruit remodelling complexes that could indirectly increase access. However, we now present evidence that non-coding transcription functions in a previously undescribed way to trigger V(D)J recombination. By directly measuring RSS accessibility, we show that transcription is essential to generate accessibility at RSSs (Figure 5) and, moreover, that it can generate this accessibility via the eviction of H2A/H2B dimers (Figures 6

and 7), a chromatin change not previously associated with V(D)J recombination.

This process transiently exposes 35–40 bp of DNA, thus increasing the probability of the RSS being available for RAG cutting. H2A/H2B eviction occurs with a displacement time of ~6 min *in vivo* (Kimura and Cook, 2001); this fits well with our observed transient accessibility of the RSSs. We also suggest that the transient nature of this eviction is well suited to the regulation of V(D)J recombination: accessibility of the RSSs for a protracted period risks too much RAG cutting and genome instability, whereas too little accessibility would impinge on the efficiency of the recombination reaction. It also implies that only a limited number of RSSs will become available for recombination at any one time, thus reducing the risk of RAG cutting at multiple sites. The transient eviction also fits well with the requirement for the stochastic use of different RSSs: each RSS has a chance of being accessible and bound by RAGs but this opportunity is short-lived. Also consistent with the stochastic nature of recombination, nucleosome mapping studies showed that RSSs do not occupy the same translational position on the nucleosome at either the $J\kappa$ or $J\lambda_1$ regions, implying that different RSSs have the potential to become accessible in different cells.

The eviction of H2A/H2B dimers during transcription elongation is facilitated by the complex FACT (Belotserkovskaya *et al*, 2003). Unfortunately, however, it was not possible to test if FACT is essential to generate RSS accessibility since its knockdown results in a depletion of all four core histones by 40–50% (Stanlie *et al*, 2010), which increases accessibility independent of any role FACT may have in H2A/H2B eviction.

Recent studies by Krangel and colleagues suggested that transcription increases RSS accessibility by repositioning nucleosomes as well as by evicting entire nucleosome cores (Kondilis-Mangum *et al*, 2010). However, whole-genome studies suggest that transcription changes nucleosome positions by only about 10 bp (Weiner *et al*, 2010) and nucleosomes are evicted only during very high levels of transcription such as that found at rRNA genes and very highly transcribed polymerase II genes (Thiriet and Hayes, 2006; Bintu *et al*, 2011). Indeed, exchange of H3, as would occur upon whole-nucleosome eviction, occurs at only about 1/20th the rate of H2A/H2B exchange (Kimura and Cook, 2001; Thiriet and Hayes, 2005; Deal *et al*, 2010), and nucleosome mapping studies indicate that RNA polymerase II does not cause substantial nucleosome displacement *in vivo* (Chodavarapu *et al*, 2010; Deal *et al*, 2010; Weiner *et al*, 2010). Instead, we suggest that at least some of the increased accessibility observed by Kondilis-Mangum *et al* (2010) could be explained by the eviction of H2A/H2B dimers. However, we cannot exclude that transcription-dependent whole-nucleosome eviction also takes place, which would further contribute to RSS accessibility. Moreover, this eviction may depend on nucleosome remodelling complexes such as SWI/SNF as suggested by previous studies (Du *et al*, 2008).

Consistent with the idea that H2A/H2B eviction plays an important role in generating RSS accessibility, the level of accessibility measured in our studies (up to 18%) fits well with the predicted accessibility from eviction of an H2A/H2B dimer. Specifically, since the heptamer plus part of the spacer

need to be completely free from histones for full RAG cutting, at least under the uncoupled cleavage conditions we used *in vitro*, then when the heptamer lies anywhere within the last 30 bp of nucleosomal DNA, hexasome formation will result in full RAG cutting. This is equivalent to about 20% of nucleosomal DNA (30 bp/150 bp) and is very close to our measured level of accessibility of 18%. In contrast, if whole nucleosomes are evicted, then most of the RSSs would be expected to be cut.

Together, our data underpin the central role of non-coding transcription in regulating locus activation for V(D)J recombination. We find that two key chromatin changes, H3K4me3 (Figure 4B) and H2A/H2B eviction (Figure 7), are directly regulated by non-coding transcription. While the former lies predominantly at the *J* gene segments (Ji *et al*, 2010b) and recruits RAG proteins (Liu *et al*, 2007; Matthews *et al*, 2007), we suggest that H2A/H2B eviction by passage of RNA polymerase II transiently and stochastically generates RSS accessibility (Figure 7) and that this is an important step that allows pre-bound RAGs access to the RSSs to initiate V(D)J recombination.

# Materials and methods

### Construct
The λ5 cassette was a gift from Niall Dillon (Sabbattini *et al*, 1999). *Irf4* cDNA was cloned into the Hinc II site of pBluescriptSK($+$/$-$) and directionally cloned into the Cla I and Xho I sites of the *λ5* cassette.

### Cell culture and purification
Pro-B and pre-B cells were cultured and purified by flow cytometry as described (Grange and Boyes, 2007); the cells were then incubated in pro-B-cell media for 1 h at 33°C, 5% $CO_2$ before processing. Primary pro-B-cell cultures were used as it is not possible to isolate pro-B cells that overexpress IRF4 directly from the bone marrow of the PIP mice. This is because increased IRF4 levels cause pro-B cells to migrate away from stromal cells, triggering their differentiation (Johnson *et al*, 2008). *Drosophila melanogaster* SL2 cells were cultured in Schneider's *Drosophila* medium (Invitrogen), with 8% fetal calf serum, 50 U/ml penicillin and 50 μg/ml streptomycin at 26°C. 103/BCL-2 cells were grown in RPMI, 10% fetal calf serum, 50 U/ml penicillin, 50 μg/ml streptomycin and 50 μM β-mercaptoethanol at 33°C.

### Analysis of Igκ cell surface expression
Cultured pro-B cells were stained as described (Grange and Boyes, 2007) with PE anti-mouse Igκ (BD Pharmingen 559940), FITC anti-mouse Ly51 (BD Pharmingen 553160) or PE anti-mouse CD43 (BD Pharmingen 553271), and analysed using a FACsCalibur (Becton Dickinson).

### Preparation of RNA and reverse transcription
Total RNA was isolated using Trizol® (Invitrogen) and reverse transcribed using M-MLV reverse transcriptase (Invitrogen), according to the manufacturer's instructions.

### Southern blotting
For semiquantitative PCR and Southern blotting, cells were lysed in 10 mM Tris pH 8.0, 1 mM EDTA, 100 mM NaCl, 1% SDS and 0.4 mg/ml proteinase K at 56°C, and the DNA was recovered. The primers and PCR conditions are given in Supplementary Table S1. Southern blotting was carried out as described (Baumann *et al*, 2003). Hybridisation was at 65°C using $^{32}$P-radiolabelled probes that had been prepared via PCR using the primers given in Supplementary Table S1. The probe to detect *Igκ* recombination hybridises to the *Jκ1* gene segment; since this region is removed upon recombination to downstream gene segments, only recombination to *Jκ1* is detected.

The copy number was determined by digesting 10 μg of kidney DNA with 40 U Pvu II before separating on a 1% agarose gel and Southern blotting. The hybridisation probe was made via PCR using the primers CNF: 5′-GCAAGTGTTTGCTCACCATGGC-3′ and CNR: 5′-GCTTCCTCTGTCTCTGAGGG-3′.

Nucleosome positions adjacent to the κ3′E and at the *Igκ* RSSs were mapped according to (Boyes and Felsenfeld, 1996) using the 103/BCL-2 cell line; this was induced to undergo IgL recombination by temperature shift from 33 to 39.7°C (Chen *et al*, 1994). Forty-eight microgram of DNA was digested with 0–45 units of micrococcal nuclease (Worthington). The DNA was then digested with Apa I/Xho I (adjacent to κ3′E) or Eco RI/Bgl II (*Jκ* region) before separation. After Southern blotting, the membrane was hybridised to a probe made via PCR using the primers: k3′EprobeF: 5′-GACAT AGCAGTGAAACATTGATGACC-3′ and k3′EprobeR: 5′-CAGAGTCTA CCGTTTCATCTCC-3′ or JkprobeF: 5′-GGGTTCTACAGGCCTAACA ACC-3′ and JkprobeR: 5′-GGAATTGAGTTATGGATACAACAGCAACC-3′.

### Western blotting
Cells were lysed in a 3:1 mix of RIPA (25 mM Tris pH 8.2, 50 mM NaCl, 0.5% NP40, 0.5% sodium deoxycholate and 0.1% sodium dodecyl sulphate) and lysis buffer (5% sodium dodecyl sulphate, 0.15 M Tris pH 6.7 and 30% glycerol) at a concentration of $2 \times 10^4$ cells/μl in the presence of protease inhibitors (Complete Inhibitor Cocktail, Roche) and the proteasome inhibitor, MG132 (2.5 μM), before boiling for 5 min. Samples were separated on a 10% acrylamide gel, transferred to polyvinylidine fluoride (PVDF) membrane (Millipore Immobilin-P) for 1 h at 0.68 mA/cm$^2$, blocked for 1 h with 5% non-fat dried milk (Marvel), in 1 × TBS (25 mM Tris, pH 7.5, 144 mM NaCl and 5 mM KCl) plus 0.05% Tween and then incubated with either anti-β-Tubulin (Sigma T4026) or anti-IRF4 (Santa Cruz sc 6059). Dilutions were according to the manufacturer's instructions.

### Chromatin immunoprecipitation
For carrier chromatin immunoprecipitation, $1 \times 10^7$ cells were mixed with $5 \times 10^7$ *Drosophila* SL2 cells and immunoprecipitation was carried out as described (O'Neill *et al*, 2006), except that mono, di and tri nucleosomes were purified on a 10–40% sucrose gradient prior to immunoprecipitation with either a pan-anti-acetyl-H4 antibody (Upstate 06–866), anti-H3K4me3 antibody (Abcam ab1012), anti-acetyl H3 antibody (Upstate 06–599), anti-H3K36me3 antibody (Abcam ab9050) or anti-IgG antibody (Upstate 12–370). Cross-linked chromatin immunoprecipitation was carried out according to (Boyd and Farnham, 1999) using anti-H2B (Abcam ab1790), anti-H2A (Abcam ab18255), anti-H3 (Abcam ab1791), anti-RNA polymerase II phosphorylated at serine 2 of the C-terminal repeat (Abcam ab5095), anti-IRF4 (Santa Cruz sc 6059) or anti-STAT5 (Abcam ab7969) antibodies. RAG1 chromatin immunoprecipitation was carried out as described (Ji *et al*, 2010b) using an anti-RAG1 antibody that was a kind gift of Prof D Schatz (Subrahmanyam *et al*, 2012). The primers used for qPCR analysis are given in Supplementary Table S2.

### Reconstitution of nucleosomes and hexasomes
Fragment 1 (146 bp) was excised from FM210 (McBlane and Boyes, 2000) with *Bam* HI and *Hin*d III. The functional 12-spacer RSS was amplified from pFM210 using the primers NRBamF: 5′-CGGATCCGC GTTCTGAACAAATCCAG-3′ and NRHindR: 5′-CCAAGCTTGCTATGA CCATGATTACGC-3′. This was cloned into the Bam HI/Eco RV site of pBluescript to generate pBsRSS; fragment 3 (158 bp) was excised using Xho I and Spe I. Fragment 2 (124 bp) was excised from pBsRSS with Sac I and Hpa II. The DNA fragments were end-labelled and reconstituted with recombinant *Xenopus* histone octamers via salt dialysis (Luger *et al*, 1997). Nucleosomes and hexasomes were gel purified and their positions mapped via micrococcal nuclease digestion as described (Baumann *et al*, 2003). RAG cutting reactions were performed as described (Baumann *et al*, 2003) using 20 fmol of each reconstitute. Restriction digestions were performed with 20 fmol of reconstitute in Sal I digestion buffer (NEB) and 15 units of Sal I at 37°C for 100 min.

### Restriction enzyme accessibility
$3 \times 10^6$ cells were resuspended in 3 ml pro-B cell media; α-amanitin (2.5 μM) was added to $1.5 \times 10^6$ cells and the cells were cultured for

1 h at 33°C, 5% $CO_2$. Nuclei were prepared as described (Grange and Boyes, 2007) and incubated at 37°C with or without 50 U of enzyme per $1 \times 10^6$ cells in a final volume of 200 μl for 20 min. Digestion was stopped by the addition of 10 mM EDTA. The level of accessibility was detected by qPCR, using the primer pairs listed in Supplementary Table S2. Accessibility was calculated from the equation $X - Y/X$, where X and Y are the amounts of non-digested DNA and digested DNA, respectively.

### Supplementary data

Supplementary data are available at *The EMBO Journal* Online (http://www.embojournal.org).

# Acknowledgements

We thank Niall Dillon for the λ5 LCR cassette, Andrew Flaus for recombinant *Xenopus* histone expression constructs and advice

regarding octamer preparation, David Schatz for the anti-RAG1 antibody, Michael Atchison for the IRF4 expression construct and Graham Bottley/Gareth Howell for flow cytometry. We are very grateful to Matthias Merkenschlager for helpful comments on the manuscript, to Adamantios Mamais, Serene Yeo and Nikoletta Pechlivani for technical help and to Chris Kirkham for an aliquot of purified RAG1. This work was funded by MRC grants G9900193 and G0801101 (to JB), and a BBSRC strategic studentship (to SB). JB gratefully acknowledges support from the Lister Institute of Preventive Medicine and a Fellowship from the University of Leeds.

*Author contributions*: SB performed the experiments, analysed the data and wrote the manuscript; JB performed the experiments for Figures 6 and 7B–E, analysed the data and wrote the manuscript.

# Conflict of interest

The authors declare that they have no conflict of interest.

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
