## [Review Process File · The EMBO Journal]

Manuscript EMBO-2012-80768

Transcription-coupled Eviction of histones H2A/H2B governs V(D)J Recombination

Sarah Bevington and Joan Boyes

Corresponding author: Joan Boyes, Institute of Molecular and Cellular Biology

Review timeline:

Submission date:	14 January 2012
Editorial Decision:	11 March 2012
Additional correspondence (author):	30 May 2012
Additional correspondence (editor):	31 May 2012
Revision received:	29 July 2012
Editorial Decision:	10 September 2012
Revision received:	25 October 2012
Editorial Decision:	18 November 2012
Revision received:	19 January 2013
Accepted	5 February 2013

Transaction Report:

Editor: Bernd Pulverer

1st Editorial Decision

11 March 2012

Thank you very much for submitting your interesting manuscript for consideration by the EMBO Journal. I apologize for the delay in sending this decision, which is based on the two referee reports shown below.

Both referees underline the potential novelty of your findings, in particular the following three aspects:

- > IRF4 activation of IgL recombination in pro-B cells,
- > histone hexamers are more permissive for RAG-mediated cleavage than histone octamers,
- > germline transcription activates V(D)J recombination through transient eviction of H2A/H2B dimers.

Unfortunately, the referees have identified multiple substantial problems with the data, the strength of the conclusions and indeed the discussion of the previous literature.

In particular, the referees are intrigued by the transient H2A/H2B eviction hypothesis, but they both note that this would have to be validated in vivo to be compelling. In particular the second referee argues that there is a lack of concrete evidence for this hypothesis.

While we appreciate that this is not an easy issue to address in revision, we nevertheless agree with the referees that this is essential as it forms a core part of the novel conclusions drawn and the referees provide specific and realistic experimental suggestions to address this point (e.g. ref 1, point 2). Assuming no related study is published in the meantime, we will of course allow the required time for revision to carry out these experiments.

Should you be able to address the referee criticisms in full- in this case we have decided that all the requested experimental revision is required - we would be very interested to consider a revised manuscript. However, please note that it is our policy to allow a single substantial round of revision only and that, therefore, acceptance or rejection of the manuscript will depend on the completeness of your responses in this revised version. I do realize that addressing all the referees' criticisms will require significant additional time and effort and is technically challenging, but we hope that you are prepared to undertake this revision.

If you decide to submit a revised manuscript, please include a detailed point-by-point response to the referees' comments. Please bear in mind that this would form part of the Review Process File upon acceptance, and will therefore be available online to the community. For more details on our Transparent Editorial Process, please visit our website:
<http://www.nature.com/emboj/about/process.html>

We generally allow three months as standard revision time. As a matter of policy, competing manuscripts published during this period will NOT negatively impact on our assessment of the conceptual advance presented by your study. However, we request that you contact the editor as soon as possible upon publication of any related work, to discuss how to proceed. As I noted, we are happy to extend the revision period as required, assuming no competing manuscript is published in the meantime - please contact me after the three month period to discuss progress.

Thank you for the opportunity to consider your potentially exciting work for publication. I look forward to your revision.

REFeree COMMENTS

Referee #1

The article by Bevington et. al. describes a novel mechanism for recombination for VD(J) recombination that is mediated by transcription mediated eviction of H2A/H2B dimer at the recombination loci. Using pro-B cells that express increased levels of IRF4, the authors show that there is increased non-coding transcription at the I_gk and I_gl loci, leading to increased RSS accessibility, which in turn is sufficient for RAG mediated cleavage of DNA. They use an in vitro model system of reassembled hexamers versus octamers to show that H2B/H2B dimer loss is sufficient for exposing 35-40 bp DNA, which in turn exposes sites for RAG cutting. The authors present a good study to substantiate their hypothesis, however a some important experiments to support their conclusions are needed, as listed below.

1. The authors show that increased transcription of non-coding transcripts is the possible reason for increased accessibility of sites for RAG1 mediated transcription. The authors should carry out ChIP analysis to show the recruitment of Pol2 at these loci, +/- a-aminin, in order to prove that transcription is the cause for increased RAG1 mediated cleavage.
2. The authors show using an in vitro system that fragments of DNA containing the RSS, assembled on hexamers or octamers are differentially cleaved by RAG1 and thus suggest that loss of H2A/H2B dimer the cause of increased recombination. The authors need to show this affect in vivo, using ChIP to show the loss of H2B/H2A at the RSS as compared to levels of H3 or H4, +/- a-aminin. Also they show should the recruitment of RAG proteins to the RSS in a transcription dependent manner.
3. The authors show attempt to show the possible mechanism by which H2A/H2B dimer is lost. IS passage of Pol2 sufficient or does it require other factors such as FACT?
4. Figure 7B is not very clear. A better figure for Mnase digestion should be provided.
5. The authors show changes in RNA levels in several figures. Error bars should be provided for changes in RNA levels as show in these figures. Also it is conventional to represent ChIP data as % Input. The authors may consider expressing their ChIP data as % Input instead as fold change.

Referee #2

This manuscript examines the question of how RSSs are rendered accessible for binding and cleavage by RAG1/RAG2. It starts with the interesting finding that elevated expression of IRF4 from a transgene results in increased recombination of the Ig kappa and Ig lambda loci in pro-B cells. The authors examine levels of histone modifications and germline transcription at the light chain loci, which reveals that the increase in light chain gene recombination correlates well with elevated levels of germline transcription, and less well with levels of histone acetylation and H3K4me3. What might germline transcription do to enhance recombination? Transcription is known to lead to the transient eviction of an H2A/H2B dimer, and the authors hypothesize that this is an important event in allowing RSS accessibility and recombination. To support this, they demonstrate that when fragments of DNA containing RSSs are assembled into histone hexamers (which lack an H2A/H2B dimer), less of the DNA is covered by the histone proteins than with a full histone octamer, as expected, and that depending on the exact location of the RSS on the DNA, a hexamer can allow more cleavage by RAG or a restriction enzyme than does an octamer. This nice biochemical result indicates that the authors' model is plausible.

The strengths of this manuscript are the novel finding concerning IRF4 activation of IgL recombination in pro-B cells, the demonstration that histone hexamers are more permissive for RAG-mediated cleavage than histone octamers, and the interesting hypothesis that germline transcription activates V(D)J recombination through transient eviction of H2A/H2B dimers. One weakness is that no *in vivo* evidence is provided for the hypothesis. There are some other weaknesses, as detailed below, a major one being that the conclusions stated in the manuscript go beyond (and in some cases, well beyond) the data. In general, there is a lack of rigor in the interpretation of the data. There are also some concerns with the experiments themselves.

Specific comments:

1. The authors write the manuscript as if their data demonstrate conclusively that the role of germline transcription in activating V(D)J recombination is to evict an H2A/H2B dimer. While this is an appealing model and is rendered plausible by their biochemical analysis, no *in vivo* support is provided for it. Transcription is a complex process involving many factors and processes, including chromatin remodeling complexes, and H2A/H2B dimer eviction might be irrelevant to V(D)J recombination *in vivo* or might play only a minor role. No data in this paper argue otherwise. There are numerous places in the manuscript where the conclusions are stated inappropriately and need to be rewritten to make it clear that while the data support the model, the model has yet to be tested or confirmed *in vivo*.
2. This issue is highlighted by the authors' treatment of previous work on the role of chromatin remodeling complexes in rendering RSSs accessible. Elegant work from the Oltz and Sen labs *in vivo* and *in vitro* have built a strong case for a role of SWI/SNF in rendering RSSs accessible for V(D)J recombination (the Sen paper is not cited and should be: Du, H. et al (2008). Activation of 12/23-RSS-dependent RAG cleavage by hSWI/SNF complex in the absence of transcription. *Mol Cell* 31, 641-649.) The evidence in favor of this model is stronger than that provided here for H2A/H2B eviction. On page 4, bottom of the first paragraph, the authors address this work by saying that SWI/SNF "is required to activate non-coding transcription *in vivo* (Osipovich et al., 2007; Osipovich et al., 2009) rather than remodelling nucleosomes at RSSs." This is not an accurate interpretation of these papers: the data of the papers are fully compatible with a role for SWI/SNF in removing/remodeling nucleosomes to allow for V(D)J recombination, but Oltz and colleagues are careful to point out that their data don't rule out a role in activating germline transcription. The elegant biochemistry in the Sen paper further emphasizes this point, which was also made in an early paper from the Oettinger lab (Kwon, J. et al. (2000). Histone acetylation and hSWI/SNF remodeling act in concert to stimulate V(D)J cleavage of nucleosomal DNA. *Mol Cell* 6, 1037-1048.). The authors need to integrate their findings with this previous work appropriately: it seems perfectly reasonable that both mechanisms (nucleosome repositioning and H2A/H2B eviction) are operating-this does not detract from the significance of the findings presented here.

3. Page 4: the authors refer to "small pre-B II and pre-B cell stages." What is the difference between small pre-B II and pre-B stages? Aren't they essentially names for the same thing?
4. Figure 2A: the only data shown for kappa recombination is V-to-Jk5, which is a strange choice since most initial recombination events focus on Jk1 (and to some extent Jk2). I'm concerned that the analysis does not accurately reflect levels of kappa recombination. Analysis of recombination to Jk1 should be performed.
5. Page 9, middle, the authors discuss the fact that high H3K4me3 levels at Jkappa in PIP3 pro-B cells don't result in full pre-B levels of recombination, and go on later in the manuscript to infer from this that reduced recombination is because germline transcripts are only partially activated. They fail to consider other possibilities, however. First, one worrisome possibility is that transgenic expression IRF4 alters the kinetics of B cell development, perhaps shortening the time spent in the pro-B compartment-this could limit kappa recombination for reasons unrelated to reduced levels of germline transcription. A second possibility not considered is that kappa recombination is reduced due to a failure of full locus contraction/looping, which again would mean that reduced recombination could be unrelated to reduced germline transcription.
6. In the same paragraph, the authors write: " This indicates that H3K4me3 and recombination are not tightly linked and, since H3K4me3 causes the recruitment of RAG2 (Ji et al., 2010) these data imply that RAG2 recruitment is not the key activating step in V(D)J recombination." This is not an accurate way to state the conclusion: strong RAG2 recruitment via H3K4me3 might be essential for activating kappa recombination-what the authors can conclude (with the caveats of point #5) is that it is not sufficient for full kappa recombination. They certainly cannot conclude that strong RAG2 recruitment is not one of the key activating steps.
7. The authors report that there is only a small (2x) increase in H3 acetylation at Jkappa between pro and pre B cells. This same point was demonstrated previously by the Bergman lab: Goldmit, M. et al. (2005). Epigenetic ontogeny of the Igk locus during B cell development. Nat Immunol 6, 198-203. This should be acknowledged.
8. In the same paragraph, the authors state: " this is compatible with previous studies that showed histone H3 hyperacetylation is insufficient for RAG cutting at heavy chain loci (Hesslein et al., 2003)." The cited study did not measure RAG cutting, only V(D)J recombination, and citing it here is misleading since the reason for the discrepancy between H3 acetylation and recombination was probably a failure of locus contraction/looping, not a failure of RAGs to bind.
9. In the same paragraph, the authors state: " Therefore, neither H4 nor H3 acetylation appear to be the critical chromatin change that increases RSS accessibility but instead, acetylation at RSSs might be a pre-requisite for other chromatin changes." This suffers from the same flaw in logic as point #6: histone acetylation might very well be one of the critical changes directly required for accessibility; the best the authors can conclude is that it is not sufficient for accessibility.
10. Page 13: in describing the in vitro cleavage reaction, the authors should explain that the assay is done in the presence of Mn⁺⁺, which allows for uncoupled (single RSS) cleavage, hence eliminating the requirement for synaptic complex formation for cleavage, which almost certainly alters the sequence requirements for cleavage (for example, probably reducing reliance on the nonamer). Hence, the cleavage data likely do not reflect what would take place on similar substrates with only Mg⁺⁺ present (as is the case in vivo). The authors extrapolate from their data in misleading ways, for example in the discussion stating "Specifically, since the heptamer plus part of the spacer need to be completely free from histones for full RAG cutting, then when the heptamer lies anywhere within the last 30 bp of nucleosomal DNA, hexasome formation will result in full RAG cutting." Such a claim might be true in Mn⁺⁺ in vitro and might very well not be true in vivo.
11. Page 15, eight lines up from bottom-a typo (extra word "binding").
12. Discussion, second paragraph: the authors refer to transcriptional enhancers as "recombination enhancers": is there evidence that Elambda3-1 is indeed a recombination enhancer?

13. Bottom of page 19, top of page 20: this paragraph contains two statements that are unjustified, as explained above. The first is the claim that chromatin modifications do not directly increase RAG access; the second is this statement: " and, moreover, that it generates this accessibility via the eviction of H2A/H2B dimers . . . "

14. Fig. 7B: this analysis is quite crude and does not support the conclusions the authors attribute to it (for example, page 20, 8-9 lines up from bottom). It does seem likely that nucleosomes are not strongly phased throughout the entire region analyzed; however, the authors would need to perform a more detailed and focused analysis (perhaps as was done by Kondilis-Mangum et al., 2010) to draw conclusions about the nucleosomes over the J gene segments.

15. Final paragraph of discussion: " We find that two key chromatin changes, H3K4me3 and H2A/H2B eviction are directly regulated by non-coding transcription." This goes far beyond the data: the authors do not do any experiments to link transcription to H2A/H2B eviction; in fact, they never measure H2A/H2B eviction at all.

Additional correspondence (author)

30 May 2012

Thank you for your letter of 11th March that invited us to submit a revised manuscript. As requested, I am contacting you after three months to discuss progress. I am grateful that you offered to extend the normal revision period since many of these experiments require primary cells from mice and it has taken some time to breed sufficient animals. However, I believe we are not too far from completing these experiments and would like to request a further six weeks to prepare everything for re-submission.

I am pleased to report that we now have in vivo data consistent with the eviction of H2A/H2B dimers at RSSs that is dependent on transcription (Reviewer 1, point 2). In four independent experiments, we find H2B is depleted by an average of 25% compared to H3 at the J κ RSSs. This same depletion is not observed when α -amanitin is added to inhibit transcription nor at an intergenic region, that is not known to be transcribed (Figure 1). Moreover, this level of H2B depletion is consistent with the level of RSS accessibility that we observed (please see calculation below).

Maximum depletion expected if a H2A/H2B dimer is lost at this site in all cells = 50%. Maximum accessibility expected if a dimer is lost at this site in all cells = $40/150 \times 100 = 27\%$ Observed level of accessibility = 17% This is 62% of the maximum accessibility 62% of maximum depletion = 32% Observed average depletion = 25% This small discrepancy could be due to accessibility being measured over time whereas depletion is a snap-shot when the cells are cross-linked.

We have also shown by ChIP that addition of α -amanitin results in loss of RNA polymerase II from the J κ RSSs (Reviewer 1, point 1) and that this reduction is to a similar level as that found in a region that is not known to be transcribed (Figure 2). These data are therefore consistent with the idea that transcription by RNA polymerase II is needed to trigger eviction of H2A/B dimers to make the RSSs accessible.

In addition, we have begun to investigate the role of transcription in the recruitment of RAG proteins, (Reviewer 1, point 2). It has been shown previously that RAG2 is bound at many sites in the genome that are enriched in H3K4me3, including sites outside the antigen receptor loci (Ji et al., 2010, Cell 141: 419-31). Consequently, we chose not to examine the role of transcription in the recruitment of RAG2. In contrast, studies using knock-out mice with deletions of recombination promoters or enhancers inferred that RAG1 is recruited to the RSSs in a transcription-dependent manner (Ji et al., 2010, J. Exp Med, 207: 2809-16); this recruitment has been proposed to depend on the accessibility of the RSSs (Ji et al., Cell, 2010). We therefore are performing ChIP of RAG1 in the presence and absence of α -amanitin and are examining RAG1 binding to the J κ RSSs where it has previously been shown to increase between the pro- and pre-B cell stages (Ji et al., Cell 2010).

Whilst the above data imply that passage of RNA polymerase II is required for eviction of H2A/H2B dimers, they do not show if it is sufficient. Reviewer 1 requested that we attempt to test if FACT is also required. However, previous studies have shown that knockdown of FACT results in

nearly 50% of all histones being depleted (Stanlie et al., 2010, PNAS 107, 22190-22195). Thus, knock-down of FACT is likely to result in increased RSS accessibility, independent of any role FACT plays in H2A/B eviction. To test this, we are in the process of performing FACT KD in the inducible pre-B cell line, 103/BCL-2. If we see increased accessibility at the J κ RSSs when FACT is knocked-down, this will imply that it is not possible to test the role of FACT in the way the reviewer suggested. If, however, there is no change in the accessibility in the absence of FACT, then we can test if this complex is essential to generate RSS accessibility. In this latter case, we aim to perform FACT knockdown in primary pro-B cells from the PIP3 transgenic mice, confirm knock-down by analysing the level of Ssrp RNA and examine the effects of FACT knock-down on RSS accessibility.

We have repeated Figure 7B to improve its quality (Reviewer 1, point 4 and Reviewer 2, point 14) and have included a control Southern blot of a region where we expect nucleosomes to be positioned. Specifically, promoters and enhancers have been shown to act as barriers that cause adjacent nucleosomes to be positioned (Mavrich et al., 2008, Genome Research 18, 1073-1083 and references therein). We therefore examined the region adjacent to the immunoglobulin kappa 3' enhancer, κ 3'E by indirect end labelling and find a clear nucleosome ladder, indicative of nucleosome positioning. By contrast, significantly more smearing is observed at the J κ RSSs, consistent with little, if any nucleosome positioning in this region (Figure 3).

Finally, as requested by reviewer 2, point 4, we have analysed recombination to J κ 1 and find exactly the same as for recombination to J κ 5 (Figure 4). Moreover, to examine Ig κ recombination in the absence of the requirement for locus contraction, we analyzed recombination between V κ 21 and J κ 1 that only about 1 kb apart. We find recombination between V κ 21 and J κ 1 is at the same level as observed for other V regions that do require locus contraction. This suggests that the reduced level of Ig κ recombination cannot be explained by a failure of locus contraction (Reviewer 2, point 5).

As far as we can see, these are all the experimental revisions that were requested; the remaining revisions require changes or clarifications to the text.

Thus, all the experimental revisions to date support our hypothesis. We request six additional weeks to complete the RAG1 ChIP experiments and to complete the FACT knockdown studies.

Thank you for considering this request.

Additional correspondence (editor)

31 May 2012

Thank you for the update on your experimental revision. We have gone through the points and agree that you are in principle addressing all the experimental issues raised (it is of course up to the referees to judge the experimental data). We understand the potential caveat about FACT and acknowledge that it makes sense to attempt the exploratory experiment suggested.

We have also checked the literature and the conceptual advance of your findings appears to stand. A further extension to the revision is therefore not a problem. Please do update us if a competing paper is published in the meantime, so that we can discuss how to proceed without delay.

1st Revision - authors' response

29 July 2012

Reviewer 1:

- 1) We have carried out an RNA polymerase II ChIP in the presence and absence of α -amanitin, using an antibody against RNA polymerase, phosphorylated at serine 2 of the CTD. Consistent with the idea that transcription elongation is required for increased RAG-mediated cleavage, we see increased RNA polymerase II binding in the absence of α -amanitin and this correlates with increased RAG1 binding and decreased levels of

H2A/H2B (point 2 below), exactly as predicted by our hypothesis. These changes are lost in the presence of a-amanitin. Binding of RNA polymerase II was examined at the *Jk* RSSs, which rearrange frequently in pre-B cells and importantly, the ChIP primers were designed so that the upstream primer for *Jk1* is within the region that is deleted upon recombination. Thus, association of RNA polymerase II due to non-coding transcription was measured rather than that due to transcription of the rearranged *Igk* gene. Although association of RNA polymerase II does not fall to exactly the same level as at a non-transcribed region in the presence of a-amanitin (as for RAG1, below), we suggest that this is because a-amanitin halts progression of the RNA polymerase II complex but does not necessarily cause it to dissociate from DNA. These data have been added as the new Figure 7C and described in the text (p15).

- 2) We have analysed the level of H2B *in vivo* compared to H3 by performing ChIP in primary pre-B cells and observe the expected depletion of H2B compared to H3 in the transcribed *Jk* region analysed above. Importantly, we do not see this same H2B depletion when transcription is inhibited by a-amanitin, nor is this depletion observed at a region that is not known to be transcribed. Likewise, we observed a similar loss of H2A in the absence of a-amanitin compared to the presence of a-amanitin in the transcribed *Jk* region but not at a non-transcribed region. Since the H2A/H2B eviction is only transient with a half-life of six minutes, the difference was not expected to be large; indeed, the maximum depletion if a H2A/H2B dimer was evicted at this site in all the cells at the time of cross-linking would be 50%. Thus, an average depletion of 25% is within the expected range and correlates well with an average RSS accessibility of 18%. (Maximum accessibility if H2A/H2B dimer missing in every cell = $40/150 = 27\%$. Thus, 18% accessibility correlates with about 33% H2A/H2B depletion). The ChIP data have been added as the new Figure 7B, Supplementary Figure S8 and described in the text p14-15.

We also examined the recruitment of RAG proteins and find that, exactly as predicted by our hypothesis, the binding of RAG1 is substantially increased by ongoing transcription. It has been shown previously that RAG2 is bound at numerous sites in the genome that are enriched in H3K4me3 (Ji et al., 2010 Cell, 141: 419-31). Thus, we did not expect the binding of RAG2 to be tightly linked to ongoing transcription. Instead, RAG1 binding has been proposed to be transcription-dependent (Ji et al., 2010 J. Exp. Med. 207: 2809-16); we therefore examined its binding at the *Jk* region and compared this to an intergenic region of the *Igl* locus. A 3-fold increase RAG1 binding was observed in the transcribed *Jk* region in the absence of a-amanitin compared to in the presence of a-amanitin. Moreover, this binding was not observed in the absence of a-amanitin at an intergenic region nor at the transcribed *Gapdh* gene that lacks RSSs. These data have been added as the new Figure 7D and described on p15.

- 3) We have attempted to investigate the mechanism by which H2A/H2B dimers are lost as requested by the reviewer. The reduced RSS accessibility, reduced association of RNA polymerase and increased presence of H2A/H2B in the presence of a-amanitin suggest that transcription elongation is indeed required for H2A/H2B eviction and RSS accessibility. To test if FACT is also required, we performed a knock-down of FACT in a pro-B cell line. Previous studies have shown that knock-down of FACT causes a substantial loss (40-50%) of all histones compared to wild type cells (Stanlie et al., 2010 PNAS 107: 22190-22195). Such histone loss is predicted to significantly increase accessibility. We tested this in our pro-B cell line and find that there is indeed a significant increase in accessibility upon FACT knock-down when transcription is ongoing (Reviewer Figure 1, at the end of this letter). Since loss of FACT correlates with a widespread increase in accessibility, it is not possible to unequivocally test if FACT plays an essential role in generating the transcription-induced RSS accessibility by accepting evicted histones. We have added a brief description to the text to clarify this (p21) and have now cited the Stanlie paper.
- 4) Figure 7B (now Figure 7E) has been repeated and a new improved Figure with additional controls has been substituted. The Figure legend has been amended accordingly (p39-40).
- 5) The changes in RNA levels are shown in Figure 3 using Southern blot data. This experiment was repeated at least three times and, as requested by the reviewer, a graph of

the quantification of these data with error bars is now additionally shown (Supplementary Figure S4).

The analyses of histone modification were done via C-ChIP due to the scarcity of primary pro- and pre-B cells. Some of the ChIP antibodies cross react with *Drosophila* histones in the carrier chromatin whereas others do not; this resulted in different amounts of immunoprecipitated material relative to input for different antibodies. To overcome this discrepancy and to more easily compare between ChIPs, we showed the level of bound material as per cent positive control. Therefore, we prefer not to change the original presentation of the ChIP data.

Reviewer 2

We have now provided *in vivo* data to support the hypothesis that non-coding transcription activates V(D)J recombination via H2A/H2B eviction as outlined above for reviewer 1, points 1 and 2. Specifically, we show that ongoing transcription is required for loss of H2A/B compared to H3 and that this increases the binding of RAG1. We have also carefully checked the text and moderated some conclusions as suggested by the reviewer and detail these below. We would like to thank the reviewer for pointing these out and hope that we have now presented a sufficiently balanced account.

- 1) As stated, we have now provided stronger *in vivo* evidence for the role of H2A/H2B eviction in V(D)J recombination and have added new Figures 7B-D and Supplementary Figure S8 to support this conclusion. We would also like to highlight that we did previously present *in vivo* evidence for this model, namely the transient nature of the accessibility (Figure 7A). Nevertheless, we have gone over the text and re-written the relevant parts of the manuscript to moderate our statements regarding the role of H2A/H2B eviction (p2 (abstract), p5, p14, p18, p21, p22 and p23) and hope that this is satisfactory.
- 2) We have now cited the paper by Du et al that provided *in vitro* evidence for the role of the hSWI/SNF complex in V(D)J recombination and re-written our description of the data from Osipovitch et al to state that their data are consistent with a role of SWI/SNF either in the initiation of non-coding transcription or in nucleosome remodeling at RSSs (p4). Moreover, as suggested by the reviewer, we have revised the text to state that nucleosome remodeling and H2A/H2B eviction might both play a role in releasing RSSs from nucleosomes for the initiation of recombination (p22).
- 3) Small pre-B II cells are the sub-set of pre-B cells where most IgL recombination occurs according to the Melchers/Rolink definition (Osmond, Melchers, Rolink, 1998, Immunology Today 19: 65-8). To avoid confusion, we have removed references to small pre-B II cells (p4 and p6) and referred to the cells as pre-B cells throughout.
- 4) In our original data, we did, in fact, analyse *Igk* recombination to *Jk1* rather than to *Jk5*. A degenerate *Vk* primer was used together with a primer that anneals downstream of *Jk5*. This amplified all of the *VkJk1-5* recombination products. However, to measure the level of *VkJk1* recombination, a Southern probe was used that hybridises to the *Jk1* gene segment. Since this region is removed upon recombination to the downstream *Jk2-5* gene segments, the Southern blot detects only *VkJk1* recombination. We have amended the text to clarify this (p25) and thank the reviewer for pointing this out.
- 5) We did, in fact, address whether IRF4 affects the kinetics of pro-B cell development and found that there was no change in pro-B cell markers over the 7 days of the experiment (Supplementary Figure 3); this implies that the time spent in the pro-B cell compartment is not shortened by the expression of IRF4. Moreover, we purified the pro-B population from the PIP3 mice to examine if increased IRF4 prematurely activates *IgL* recombination. Within the same cells, we observe *Igl* is fully activated whereas *Igk* is only partially activated. We have amended the text to clarify that increased expression of IRF4 did not seem to alter the kinetics of pro-B development (p1-2 of the Supplementary Figure legends).

We considered that *Igk* locus contraction might be the reason for the low level of *Igk* recombination and therefore analysed *Vk21/Jk1* recombination. The *Vk21* RSS lies only 1 kb from the *Jk1* RSS and thus *Vk21/Jk1* recombination is highly unlikely to require locus contraction. Nevertheless, recombination between these RSSs was also increased by only

two-fold in the presence of increased IRF4. These data have been added as the new Supplementary Figure 2A (right) and described in the text (p7).

- 6) We have amended the text as the reviewer suggested (p9, bottom) and thank the reviewer for pointing this out.
- 7) We have now referenced the Goldmit et al. paper (p11, bottom and p19) and thank the reviewer for pointing this out.
- 8) We have removed the reference for the Hesslein et al paper and again thank the reviewer.
- 9) We have re-phrased this sentence according to the reviewer's suggestions (p12, top).
- 10) We have amended the text as suggested by the reviewer to note that our *in vitro* cutting reactions were not performed under coupled cleavage conditions as would occur *in vivo* (p13). We have also highlighted this caveat when considering the amount of DNA that needs to be free of the hexasome for full cutting (p22).
- 11) We have corrected the typo and thank the reviewer for pointing this out (now p17).
- 12) We do have evidence that $El_{3,1}$ is indeed a recombination enhancer: We individually deleted the four hypersensitive sites that lie downstream of *JCI1* in BAC constructs and used these to generate transgenic mice; we found that loss of $El_{3,1}$ completely ablated recombination. This forms the basis of a further manuscript that is currently being submitted elsewhere. Since EMBO J discourages the use of the term "data not shown", we have amended the text and removed the word "recombination" (p18).
- 13) Of the various histone modifications, only histone acetylation has been demonstrated to directly increase the accessibility of any protein to its DNA binding site in chromatin. In the case of RAGs, histone acetylation was proposed to increase RAG binding to nucleosomes *in vitro* in one study (Kwon et al., 2000 Molecular Cell 6: 1037-48) but this was not found to be the case in two other studies (Golding et al., 1999, EMBO J. 18: 3712-3123 and McBlane and Boyes 2000 Current Biology 10: 483-6). Instead, histone acetylation appears to help to recruit remodeling complexes that can then increase accessibility (Nightingale et al., 2007 Nucleic Acids Research 35: 6311-21). Likewise, other histone modifications, such as H3K4me3, can also recruit nucleosome remodeling complexes to **indirectly** increase accessibility. Hence, we believe that statement that these modifications do not **directly** increase RAG access is correct. However, we have modified the text to clarify that these modifications can indirectly increase RSS accessibility via remodeling complexes (p20). We have also modulated the second statement regarding H2A/H2B eviction to highlight the new evidence that we now provide to substantiate it (p21).
- 14) We have now performed a more detailed and focused analysis of translational nucleosome positioning at the *Jk* RSSs. Translational nucleosome positioning is traditionally determined using micrococcal nuclease digestion, followed by indirect end labeling. This has the advantages that (i) nucleosome positioning can be readily observed by visualization of a Southern blot and (ii) a series of increasing micrococcal nuclease digestions are assessed and thus the experiment is less prone to artifacts due to over/under digestion as can happen with PCR-based approaches. We therefore used the Southern blot approach but to improve the quality of the Figure, we now additionally show (a) the level of micrococcal nuclease digestion of the DNA and (b) a control Southern blot to analyse nucleosome positioning adjacent to the *Igk* 3' enhancer, k3'E. Nucleosomes are expected to be positioned in this region since promoters and enhancers act as barriers that cause positioning of adjacent nucleosomes (Mavrich et al., 2008, Genome Research 18, 1073-1083 and references therein). We observe a typical nucleosome ladder in this region, indicative of nucleosome positioning. Finally, we analysed nucleosome positioning at the *Jk* RSSs and observe a more smeared pattern, indicative of little or no translational nucleosome positioning; this is consistent with our previous data. The revised Figure has been substituted and the Figure legend amended accordingly (p39-40).
- 15) In response to reviewer 1, we have now provided evidence that H2A/H2B are evicted and that addition of α -amanitin eliminates this eviction and results in decreased association of RNA polymerase II. We have therefore amended this statement to refer to the new data and hope that this is acceptable (p23).

Bevington Reviewer Figure 1

Description of Reviewer Figure 1:

The pro-B cell line 103/BCL-2 can be induced to undergo IgL recombination upon temperature shift from 33°C to 38°C, concomitant with a substantial increase in non-coding transcription (A). In cells grown at 33°C, there is a low level of accessibility at *Jκ3*, which is increased substantially upon temperature shift (B, left). When cells are transfected with siRNA against the SSRP1 subunit of FACT (siRNA+) and maintained at 33°C, there is no significant change in accessibility at *Jκ3* compared to cells that have not been transfected with siRNA (B). However, if cells are transfected with siRNA against SSRP1 (siRNA+) and temperature shifted, there is a substantial increase in accessibility over and above that observed due to temperature shift alone (B). This increased accessibility is fully consistent with previous data where knock-down of FACT results in the depletion of all four core histones by 40-50% (Stanlie et al, 2010, PNAS 107: 22190-22195). It is also consistent with increased transcription that has been observed previously when histone replenishment is lost upon mutation of *Spt6* (Kaplan et al, 2003, Science 301: 1096-1099).

Since we do not see an increase in accessibility upon knock-down of FACT in the absence of a temperature shift, this suggests that loss of FACT alone is insufficient to cause accessibility and/or histone depletion. Instead, our data are consistent with the idea that transcription triggers the displacement of histones and, in the absence of FACT, these histones are not fully replenished. However, since when transcription is ongoing, loss of FACT results in increased accessibility, it is not possible to conclusively test if FACT plays an essential role in generating the initial accessibility: The small amount of residual FACT following knock-down could be sufficient to accept the histones displaced by transcription. Only a lack of accessibility when FACT is knocked down would permit us to conclude that FACT plays an essential role in generating the accessibility.

Experimental details.

103/BCL-2 cells were grown at 33°C in RPMI supplemented with 10% foetal calf serum, 50 mM β-mercaptoethanol and 50 ug/ml streptomycin, 50 μM β-mercaptoethanol. 1×10^7 cells were transfected using the Amaxa mouse B cell nucleofector kit according to the manufacturer's instructions with 3.2 pmol siGLO indicator (Dharmacon), with and without 50 pmol stealth siRNA (Invitrogen) for SSRP1 (Stanlie et al, 2010, PNAS 107: 22190-22195, SSRP1#1). After incubation at 33°C for 16 hours, where indicated the transfected cells were temperature shifted to 38°C and maintained for 24 hours. The transfected cells were then purified by flow cytometry based on the siGLO marker; RNA was prepared using Trizol, according to the manufacturer's instructions and the accessibility at *Jκ3* was measured as described previously. Knock-down of *Ssrp1* RNA was confirmed using the primers described by Stanlie et al. (2010).

Thank you for submitting your manuscript for consideration by the EMBO Journal. It has now been seen by the two referees whose comments are enclosed.

I apologize for the slow response due to summer travel.

Referee one recommends publication and appreciates that the suggested FACT experiments were carried out.

Referee two sent an in our view highly constructive report that questions the specificity of the RAG1 and IRF4 Antibodies used. It is obviously essential to be assured of the specificity of the ChIP data. We would therefore like to request that appropriate control experiments are included, in particular the suggested ChIP in Rag1 null cells.

This important data can only be published if we are assured that the data are conclusive.

The referee also suggest a number of textural modifications and we agree that it will be important to use more conservative wording in the title and abstract.

Given the referees' positive recommendations, I would like to invite you to submit a suitably revised final version of the manuscript.

When preparing your letter of response to the referees' comments, please bear in mind that this will form part of the Review Process File, and will therefore be available online to the community. For more details on our Transparent Editorial Process, please visit our website:

<http://www.nature.com/emboj/about/process.html>

Please submit this revision within 6 weeks. As a matter of policy, competing manuscripts published during this period will not negatively impact on our assessment of the conceptual advance presented by your study. However, we request that you contact the editor as soon as possible upon publication of any related work, to discuss how to proceed.

Thank you for the opportunity to re-consider your interesting work for publication. I look forward to your revision.

REFEREE COMMENTS

Referee #1

The article by Bevington et. al. sheds new light into the mechanism of VD(J) recombination. The authors show an exciting mechanism of how recombination at these loci is mediated by eviction of H2A/H2B dimer during transcription. Using pro-B cells that express increased levels of IRF4 and pre-B cells, the authors demonstrate that increased non-coding transcription at the Igk and Igl loci, leads to H2A/H2B dimer loss in a transcription dependent manner. Using both in vivo and in vitro systems the authors show that this leads to increased RSS accessibility, recruitment of Rag proteins and subsequent Rag mediated cleavage of DNA. This article also clarifies how transcription mediated recombination is specifically due to transcription coupled loss of H2A/H2B dimer and not due to changes in histone modifications.

The authors have addressed most of the questions that were raised earlier. The FACT experiments are inconclusive, but the authors are appreciated for trying them out and mentioning them in the discussion. It would be best if better and slightly enlarged versions of Fig 7E were provided. The article however, in its current form, should be accepted for publication.

Referee #2

The revised manuscript is substantially improved with the addition of new data and improvements to the text. Overall, the data provide support for the hypothesis that germline transcription activates V(D)J recombination in part by transiently evicting H2A/H2B dimers from nucleosomes, thereby rendering RSSs more likely to be accessible for RAG binding. This is an interesting and novel hypothesis and the data in favor of it are reasonably strong. I would support publication in EMBO J. once the authors have addressed four concerns, one of which (#1) is quite serious.

1. The authors have added new RAG1 ChIP data to support the idea that RAG1 binding is enhanced by transcription. The data are not particularly strong, extensive, or well controlled, but the major problem is that the authors have used an anti-RAG1 antibody that might be junk. The antibody comes from Santa Cruz, which has a poor reputation, and in particular, sells anti-RAG1 reagents that are notoriously terrible. The antibody used by the authors, sc 363, appears to be an example of this. The product datasheet on the company website shows a western blot using sc 363 on nuclear extract of LADMAC cells-LADMAC cells are of myeloid origin (ATCC #CRL-2420) and are highly unlikely to express RAG1 at all. What the major band is in the western blot is anybody's guess, but it is unlikely to be RAG1 (RAG1 is not easy to visualize by western, so another hint that this is not RAG1 is that the band on the western is quite strong). Another comical Santa Cruz anti-RAG1 antibody is sc-5599, whose product datasheet shows staining of Jurkatt cells-a mature T cell line that doesn't express RAG1. The authors should very carefully verify the specificity of this antibody and, if they insist on using it for ChIP experiments, should perform control ChIPs on cells lacking RAG1 (preferably, RAG1^{-/-} pre-B cells). These control data should be shown, at least for the reviewers. I would add that I am not aware of any commercially available anti-RAG1 antibody that works in ChIP experiments. While it would be nice to have RAG1 ChIP experiments in the manuscript, I would much rather see the manuscript published without such experiments at all than published with the current, questionable data.

I'm also concerned that the authors have used an anti-IRF4 reagent from Santa Cruz: have they carefully validated the specificity of this antibody?

2. In the manuscript and rebuttal, the authors state that Vk21 is only 1 kb from Jk1: this is incorrect, and the authors could have figured out that it was incorrect from their own diagrams. For example, the diagram in Fig. 3A, top, shows (correctly) that there are two germline promoters upstream of Jk1 that lie between Jk1 and the first Vk. These two germline promoters lie roughly 1 and 8 kb upstream of Jk1 (I might have the exact distances wrong), which means that the nearest Vk must be more than 8 kb away. I believe the distance between Jk1 and the closest Vk is on the order of 20 kb. Obviously, this needs to be corrected in the manuscript, and it undermines the authors argument that looping/proximity isn't relevant to the interpretation of their data.

3. Figure 2A: I'm glad that the authors have explained in the Materials and Methods that the Igk recombination assay detects V-to-Jk1 products-unfortunately, most readers won't look at the M&M and will be misled, as I was, by the figure. It would be quite easy to make the figure clearer. I suggest that the authors label the PCR product as "V-to-Jk1 recombination" instead of "Igk recombination", and/or indicate the location of the probe, or failing this, at least explain in the figure legend what is being detected.

4. The authors have done a nice job, for the most part, of restating their conclusions appropriately. I remain concerned, however, about the title of the manuscript, which states unequivocally that IRF4 works via eviction of H2A/H2B dimers. I do not believe that the authors have demonstrated this with certainty or even with sufficient likelihood to warrant such a statement. I do believe that they have provided substantial support for the hypothesis, but the data as they stand provide fairly indirect support. In particular, the authors have not shown that if eviction of H2A/H2B is itself blocked (not simply by blocking transcription, which changes many things), then recombination is reduced. Nor have they shown that if H2A/H2B eviction is itself increased (not simply by increasing transcription, which changes many things), then recombination is increased. This is not the authors' fault and I'm not suggesting that they do more experiments-in fact, I'm not sure how one could do such experiments in vivo, and even in vitro they would be difficult. I would not force the authors to change the title-ultimately, it's their choice-but I am not comfortable with it because I believe it goes beyond the data.

The second statement that is questionable is in the abstract, where the authors claim "that non-coding transcription is essential for RSS accessibility". There are several papers in the field that argue that in certain cases, transcription through an RSS is NOT essential for accessibility and recombination. See, for example, Sikes, M.L., Meade, A., Tripathi, R., Krangel, M.S., and Oltz, E.M. (2002). Regulation of V(D)J recombination: a dominant role for promoter positioning in gene segment accessibility. *Proc Natl Acad Sci USA* 99, 12309-12314. Are the authors saying that these previous papers were all wrong? If so, they had better discuss this point in some detail. A better option would be to soften this statement.

2nd Revision - authors' response

25 October 2012

Reviewer 1:

We are very pleased that the reviewer recommends publication and have provided slightly enlarged versions of Figure 7E as requested.

Reviewer 2:

- 1) We thank the reviewer for pointing out the potential caveat with the anti-RAG1 antibody. We originally tested various anti-RAG1 antibodies in ChIP experiments using the pre-B cell line, 103/BCL-2, and found that the sc363 antibody immunoprecipitated DNA from the Jk1 and J11 regions but not from regions outside the RSSs. Given this positive result, we used this antibody in chromatin immunoprecipitation experiments using primary pre-B cells (Figure 7D). We have now better tested the specificity of this antibody by firstly performing a western blot using extracts from various B and non-B cell lines (63-12, 103/BCL-2, MEL and NIH3T3; Reviewer Figure 1, at the end of this letter) and secondly by performing a chromatin immunoprecipitation in *Rag1*^{-/-} B cells (new Supplementary Figure 8B). As can be seen in Reviewer Figure 1, the antibody strongly detects RAG1 that has been purified from Sf9 cells as well as a protein of the molecular weight of RAG1 in B cell lines. However, it also detects a protein of slightly higher molecular weight in MEL and NIH3T3 cells, as well as some lower molecular weight bands. Thus, whilst the antibody clearly detects RAG1, it also cross reacts with one or more widely expressed proteins.

To determine whether this cross reactivity affects our chromatin immunoprecipitation results, we performed control chromatin immunoprecipitations in *Rag1*^{-/-} cells. Previous studies by Ji et al., (2010; *Cell*, 141: 419-31) showed that RAG1 binds the Jh2 region in pro-B cells and that this binding persists in pre-B cells (Ji et al., 2010, Supplementary Figure 5); minimal binding was observed in *Rag1*^{-/-} cells. Since we were only able to obtain *Rag1*^{-/-} pro-B cells and not *Rag1*^{-/-} *Igh* pre-B cells on the time-scale of these revisions, we performed the ChIP with the sc363 antibody in the *Rag1*^{-/-} pro-B cells. As can be seen in Supplementary Figure 8B, only background levels of binding are observed at the Jh2 region in *Rag1*^{-/-} pro-B cells. However, in pre-B cells from wild type mice, where RAG1 is known to bind to the Jh2 region, we detect binding in the absence of a-amanitin but not in the presence of a-amanitin. These data therefore imply that the antibody indeed detects RAG1 binding and that the data in Figure 7D reflect specific changes in RAG1 binding in the presence and absence of a-amanitin.

It is notable, however, that the enrichment of RAG1 binding that we observe is considerably less than that reported by Ji et al (2010). Thus, whilst the antibody was sufficient to detect a transcription-dependent change in RAG1 binding at Jk1, this might not reflect the true enrichment in binding. We have therefore pointed this out in the text and moderated our conclusions accordingly (p16); we hope that this is acceptable.

We also tested the specificity of the IRF4 antibody. Using the same cell lines as above, we find that the antibody detects a single band of the correct size (51.8 kDa) in a western blot using extract from the B cell lines (63-12 and 103/BCL-2) whereas no protein is detected in extract from non-B cell lines (MEL and NIH3T3; Bevington 80768R1 Reviewer Figure 2A).

Moreover, this antibody only detects a band of the size of IRF4 in Cos 7 cells that have been transfected with an expression vector for IRF4 and not in cells transfected with an empty vector (Bevington EMBOJ R1 Reviewer Figure 2B). Together, these data strongly imply that the IRF4 antibody is specific.

- 2) We thank the reviewer for pointing out the error in the distance between Vk21 and Jk1. This is in fact 18.4 kb and we have now corrected this in the text. However, we do not believe that this undermines our argument about locus contraction; 3C analyses have shown that numerous enhancers and promoters interact over this distance by locus looping but without large scale locus contraction, for example Vernimmen et al., (EMBO J. 26, 2041-51; 2007) detected interactions over 40 kb at the α -globin locus and Tolhuis et al (Mol Cell 10, 1453-65; 2002) detected interactions over 50 kb in the β -globin locus. Moreover, V11 and Jc11 are 20 kb apart and their recombination is stimulated fully by increased expression of IRF4 (Figure 2B). We have therefore modified our discussion of locus contraction (p7) and hope that this is acceptable.
- 3) We thank the reviewer for his/her suggestions regarding the labelling of Figure 2A. We have now re-labelled the PCR product as suggested by the reviewer, added the position of the probe to the schematic and referred to this in the Figure legend (p37).
- 4) We have amended the title to “Transcription-Mediated Eviction of H2A/H2B Dimers Promotes RSS Accessibility for V(D)J Recombination” and hope that this is acceptable. Likewise, we have softened the statement in the abstract regarding the essential role of non-coding transcription in generating RSS accessibility (p2).

We believe that this addresses all of the reviewer’s concerns and hope that the manuscript is now acceptable for publication.

Bevington Reviewer Figure 1

Bevington Reviewer Figure 2

A

B

Legends to Reviewer Figures

Reviewer Figure 1: Detection of RAG1 by western blotting. Left panel: Whole cell extracts were prepared from pro- and pre-B cell lines (63-12 and 103/BCL-2) as well as from the non-B cell lines (MEL and NIH3T3); 5×10^6 cells were used in each case. Core RAG1, tagged with maltose binding protein, was purified from Sf9 cells and a diluted aliquot was used as a positive control. This has a molecular weight of 116 kDa, which is very close to the molecular weight of full length RAG1 (119 kDa). The membrane was hybridized with the anti-RAG1 antibody sc363. Right panel: Coomassie gel of tagged, core RAG1 and RAG2 purified from Sf9 cells.

Reviewer Figure 2: Detection of IRF4 in cell lines and transfected Cos 7 cells with anti-IRF4 antibody. A) Detection of IRF4 in cell lines. Whole cell extracts were prepared from the pro-B, pre-B and non-B cell lines as described for Reviewer Figure 1 above. The membrane was hybridized with the anti-IRF4 antibody (sc6059), which detects a protein of the expected size for IRF4 (51.8 kDa). **B)** Detection of IRF4 in transfected Cos 7 cells. Cos 7 cells were transfected with either vector only (CMV or myc) or with increasing amounts of the IRF4 expression vector, RccMV-IRF4. Western blotting was performed with the anti-IRF4 antibody, sc6059.

Thank you for submitting your revised manuscript for consideration by the EMBO Journal. It has now been seen by the second referee whose comments are shown below.

We have evaluated your revised data in light of the advice of this referee and I am afraid that we entirely agree with the referee that the data do not provide definitive evidence for the highly interesting claims made in the title and abstract of the manuscript.

We therefore remain unable to publish the current manuscript.

In particular, we agree with referee 2 that the MW of the 'non-specific' band is not substantially different from the 'Rag1' band (and this extends also to one of the lower MW bands). Furthermore, we agree that a detailed comparison to the Ji et al. paper further undermines the claims made in the manuscript.

The clear recommendation at this stage has to be to obtain the antibody utilized in Ji et al. and to repeat the relevant experiments with this reagent. We trust that it is possible to obtain this antibody as per standard materials sharing guidelines.

Please note that successful revision to provide compelling evidence to support the claims made in the title of your manuscript will be a precondition for publication of this potentially interesting manuscript. Please also note that we can only consider one further revision of this manuscript to avoid undue protraction of the editorial process.

When preparing your letter of response to the referees' comments, please bear in mind that this will form part of the Review Process File, and will therefore be available online to the community. For more details on our Transparent Editorial Process, please visit our website: <http://www.nature.com/emboj/about/process.html>

Thank you for the opportunity to consider your work for publication. I look forward to your revision.

REFEREE COMMENTS

Referee #2

I was concerned about the RAG1 ChIP data in the first revision of the manuscript. The authors now present control experiments in the second revision to address the specificity of the anti-RAG1 antibody used. Unfortunately, the new data have made things worse, rather than better. Whereas previously I was concerned about the validity of the data, I am now convinced that the RAG1 ChIP data are not valid. There are at least two major problems. First, in Reviewer Figure 1, left hand panel, the band at the position of RAG1 in Mel and NIH3T3 cells is at exactly the same position in the blot as in the pre-B cell lines (I don't know why the authors claim that there is a difference in their rebuttal letter). This experiment provides no evidence whatsoever that this antibody is able to recognize endogenous RAG1. The fact that there is a signal for RAG1 with the purified protein suggests that the antibody has some reactivity to RAG1, but that's the most that can be concluded (the strength of this inference depends on how much protein was loaded in the lane). Second, there

are huge problems with the new data of Fig. S8B. The authors note that the signal is considerably lower than in the previously published paper by Ji et al.: this is something of an understatement. The signal at Jh2 in pre-B cells in the absence of alpha-amanitin is 1.5 fold above background, where background is defined by the signal at Gapdh. In Ji et al., the signal at Jh2 is 50 fold or more (see below) above such a background. A signal of 1.5 fold above background is simply not convincing. Equally concerning is the control experiment done with RAG1^{-/-} cells. The authors note that they were forced by circumstance to do the control in RAG1^{-/-} pro-B cells, whereas the experiment was done in WT pre-B cells. The difference the authors report in signal between the two is perhaps two-fold: unfortunately, when one examines the published data of Ji et al., they also see a stronger signal in pre-B cells than in pro-B cells (compare Fig. 2A to Fig. S5A in Ji et al.). In fact, the difference Ji et al. see is six-fold! That is, the published data would indicate that a positive signal in pro-B cells would be six fold lower than in pre-B cells. Since the authors have such a low signal to start with in pre-B cells, there was no way they could possibly have detected a positive signal in RAG1^{+/+} pro-B cells, so the experiment in RAG1^{-/-} pro-B cells was sure to give a signal at background and is not a valid control. The authors should simply give up on this antibody: it is now convincingly demonstrated, in their own hands, to be a terrible reagent.

While the authors have made a genuine effort to address my concerns, the new data have greatly exacerbated my worries, and I am now convinced that this antibody cannot be used to generate valid RAG1 ChIP data. There is no meaningful data in the manuscript concerning the effect of transcription on RAG binding, which undermines its significance.

3rd Revision - authors' response

19 January 2013

Response to Reviewer 2 and Editor's Comments

As suggested, we requested a more specific anti-RAG1 antibody from Professor Schatz, who kindly provided antibody no. 23. Whilst this is not the same as used in the manuscript by Ji et al., (2010) due to availability issues, the specificity of the new anti-RAG1 antibody has been rigorously tested (Subrahmanyam et al., Supplementary Figure 9) and has been used in published chromatin immunoprecipitation experiments (Subrahmanyam et al., 2012, Nature Immunology 13: 1205-12).

We repeated the RAG1 chromatin immunoprecipitation experiments using this antibody in the presence and absence of a-amanitin and observe a major enrichment of RAG1 binding at the RSS when transcription is ongoing (500-fold); this is significantly reduced when transcription is inhibited. This is fully consistent with the idea that transcription makes the RSSs accessible for RAG proteins to bind and initiate V(D)J recombination and thus substantiates the claims made in our manuscript.

Whilst these data agree with our model, they also imply that the association of RAG1 with recombining loci is more complicated than RAG1/RSS binding alone: Some RAG1 remains following the inhibition of transcription and loss of RSS accessibility. This suggests that mechanisms might exist to retain RAG1. Given the transient nature of the RSS accessibility (the half life of H2A/H2B eviction is six minutes), such local retention of RAG1 could be important to ensure that RAG proteins are available and ready to initiate recombination once the RSSs do become accessible. This retention could occur via a number of mechanisms: One possibility is that it occurs via interaction of RAG1 with RAG2; the latter is recruited via interaction of its PHD domain with H3K4me3 and thus can remain at antigen receptor loci in the absence of RSS binding. Another possibility is that the low level of RSS accessibility observed in the absence of transcription permits RAG binding in a fraction of the cells. Notably, the published experiments which suggested that RAG1 binding "likely" requires interaction with the RSS were performed using DNA templates that completely lack an RSS (Ji et al., 2010 Cell 141, 419-431); thus, the retention of RAG1 at endogenous antigen receptor loci has not been previously examined.

Deciphering the exact mechanism of RAG1 retention will require very detailed analysis and is beyond the scope of this manuscript. Moreover, we would like to highlight that this, and

RAG1 binding, are **not** the focus of this work. Instead, the major emphasis of our manuscript, reflected in the title and abstract, is the mechanism by which RSSs are made accessible for the initiation recombination, namely the transient eviction of H2A/H2B. Whilst the increase in accessibility inevitably facilitates RAG binding, this is exactly what we observe and is what we present in our new data.

We have substituted this new data as the revised Figure 7D and discussed it in the text (p15-16). We have also removed Supplementary Figure 8B and its legend since this showed control experiments for the anti-RAG1 antibody used previously.

We hope that these revisions are satisfactory and that the manuscript is now acceptable for publication.

4th Editorial Decision

5 February 2013

Thank you for carrying out the final experimental revisions with the new antibody. You will see below that the referee is now entirely in favour of publication.

I am therefore very pleased to inform you that your manuscript has been accepted for publication in the EMBO Journal.

REFEREE COMMENTS

Referee #2

The use of the new anti-RAG1 antibody has addressed all of my concerns. In the RAG1 ChIP experiment of Fig. 7D, the fold enrichment at Jk1 was about 0.55 in the previous version of the manuscript and is now about 500--a 1000 fold increase. The data now make sense and nicely support the authors' model.